# Dual-comb optical activity spectroscopy for the analysis of vibrational optical activity induced by external magnetic field

Daowang Peng[1,2], Chenglin Gu[1,2], Zhong Zuo[1], Yuanfeng Di[1], Xing Zou[1], Lulu Tang[1], Lunhua Deng[1], Daping Luo[1], Yang Liu[1] & Wenxue Li[1] ✉

Optical activity (OA) spectroscopy is a powerful tool to characterize molecular chirality, explore the stereo-specific structure and study the solution-state conformation of biomolecules, which is widely utilized in the fields of molecular chirality, pharmaceutics and analytical chemistry. Due to the considerably weak effect, OA spectral analysis has high demands on measurement speed and sensitivity, especially for organic biomolecules. Moreover, gas-phase OA measurements require higher resolution to resolve Doppler-limited profiles. Here, we show the unmatched potential of dual-comb spectroscopy (DCS) in magnetic optical activity spectroscopy (MOAS) of gas-phase molecules with the resolution of hundred-MHz level and the high-speed measurement of sub-millisecond level. As a demonstration, we achieved the rapid, high-precision and high-resolution MOAS measurement of the nitrogen dioxide $v_1+v_3$ band and the nitric oxide overtone band, which can be used to analyze fine structure of molecules. Besides, the preliminary demonstration of liquid-phase chiroptical activity (as weak as $10^{-5}$) has been achieved with several seconds of sampling time, which could become a routine approach enabling ultrafast dynamics analysis of chiral structural conformations.

Optical activity (OA) phenomena reflecting different interactions of molecules with left- and right-circular polarized light have attracted much attention, motivated by the unique characterizations that can reveal abundant information, such as the absolute structural configuration of chiral molecules[1–3] and the external field response of achiral molecules[4–7]. In recent years, chiroptical activity (COA)[8], Raman optical activity (ROA)[9] and magnetic optical activity (MOA)[10] have been developed to explore the interaction of matter with polarized light. Typically, electronic/vibrational circular dichroism (E/VCD) and optical rotatory dispersion (ORD), spectrally resolving the imaginary and real parts of a complex OA susceptibility, are routinely employed to determine absolutely structural configurations about chemical and biological systems, which have been widely used in pharmacology[11], asymmetric synthesis[12], and analytical sciences[13]. In particular, the study of gas-phase optical activity provides an accurate set of spectral molecular constants to facilitate spectral feature retrieval and absolute conformation identification, which can be used to calibrate ab-initio theoretical predictions and examine the role of solute-solvent interactions[14–16].

For a long time, VCD and ORD measurements have relied on fast polarization modulation of linearly polarized monochromatic light with a photoelastic modulator (PEM). Due to differences in absorption and optical rotation measurements, commercial VCD spectrometers and ORD polarimeters are two devices with different configurations. However, the conventional spectropolarimeters have been limited to require long acquisition times for quite low scanning rate and significant artefacts introduced by optical components[17]. To overcome these problems, Cho M.[18] and Cerullo Giulio[19] have developed frequency/time-domain interferometric method based on a cross-polarization detection scheme, which detected the phase and amplitude of OA free-induction-decay (FID) field to simultaneously obtain

[1]State Key Laboratory of Precision Spectroscopy, East China Normal University, Shanghai 200062, China. [2]These authors contributed equally: Daowang Peng, Chenglin Gu. ✉e-mail: wxli@phy.ecnu.edu.cn

the broadband VCD and ORD spectra. In addition, polarization-dispersive imaging CD spectrometers based on the liquid crystal polarization grating (LCPG)[20] have tremendous potential in rapid and simple the spectral measurements of OA. Although these methods offer high sensitivity, the rotationally-resolved OA analysis of gas-phase molecules is particularly challenging due to the limited spectral resolution and long acquisition time caused by dispersive elements and mechanical scanning. Keiderling et al.[21,22] utilized broadband FT spectroscopy to study the magnetic VCD of gas-phase paramagnetic and diamagnetic molecules with a sensitivity of $10^{-4}$ (over an hour of the scanning time), and a resolution of 0.1 cm$^{-1}$. Recently, Aleksandra Foltynowicz's group introduced the optical frequency comb into Faraday rotation spectroscopy and measured the high-resolution gas-phase spectrum using a long-range scanning Fourier spectrometer[23]. The sensitivity is limited to be ~$10^{-3}$ @3500 s by the scanning speed and intrinsic noise of the Fourier spectrometer.

We focus on the emerging dual-comb spectroscopy (DCS)[24–28] to perform scanless and rapid acquisition of broadband MOA spectroscopy in gas-phase spectral analysis. Based on the asynchronous sampling of two optical combs with slightly different repetition frequencies, DCS can realize the rapid acquisition of time-domain interferogram signals without a moving mechanical stage and restore a high-resolution broadband spectrum. DCS, an emerging spectral measurement technique with advantages of the high-frequency precision of tuneable diode laser absorption spectroscopy and broadband measurement of FTS, has innovated a variety of traditional spectroscopy technologies to overcome the challenges in their widespread applications, typically such as the frequency resolution, measurement speed and spectral range. The examples are as follows: broadband coherent cavity-enhanced dual-comb spectroscopy (CE-DCS)[29,30] has been implemented to overcome the drawback of limited spectral tuning range in the traditional continuous wave cavity ring-down spectrometer (CW-CRD) to detect a variety of trace molecules simultaneously in realistic ambient conditions. Nathalie Picqué's group combines the advantages of coherent anti-Stokes Raman spectroscopy (CARS) with the resolution/accuracy of optical combs to realize the high-speed broadband CARS measurement and spectro-imaging of complex samples[31,32]. Steven T. Cundiff et al. creatively proposed tri-comb spectroscopy (TCS) to break through the limitation of resolution and acquisition speed in the measurement of multidimensional coherent spectroscopy (MDCS), which provided a comb-resolution multidimensional coherent spectrum under 1 s with decoupling homogeneous and non-homogeneous linewidths[33]. In addition, more spectroscopy techniques, such as photo-acoustic spectroscopy[34,35], spectroscopic ellipsometry[36] and laser-induced breakdown spectroscopy[37], have achieved broadband wavelength coverage, high resolution and high acquisition speed by combining with DCS, which has revolutionized precision spectroscopy.

In this paper, we report an experimental demonstration of magnetic optical activity spectroscopy with a high-frequency resolution of hundred MHz, a sensitivity of $\triangle A \sim 1.31 \times 10^{-4}$@1000 s ($\triangle\varphi \sim 0.6 \times 10^{-4}$ rad @1000 s) and high-speed broadband measurements at the sub-millisecond level by introducing DCS, namely, dual-comb optical activity spectroscopy (DC-OAS). Here, we combine the technical scheme of cross-polarization detection[18] where the broadband linearly polarized optical frequency comb (OFC) is utilized to simultaneously excite left- and right-circular absorption. Two highly coherent OFCs with broad spectral ranges were used to achieve the rapid asynchronous optical sampling of MOA response at an equal interval of 100 fs. The broadband MOA information is recorded simultaneously in a sequence of time interferograms by a single radio-frequency detector without mechanical frequency scanning. As a result, our DC-OAS technique realized the rapid measurement of broadband MOA spectra, including magnetic VCD (MVCD) and

magnetic ORD (MORD). In the experiment, the rotational-resolved MOA spectra of nitric oxide and nitrogen dioxide, covering the $\upsilon_1+\upsilon_3$ band of the nitrogen dioxide (NO$_2$) molecule ranging from 2850 to 2950 cm$^{-1}$ and part of the overtone $2\upsilon_0$ band of the nitric oxide (NO) molecule ranging from 3695 to 3775 cm$^{-1}$, were measured. The MOA information with such a broad spectral range was obtained with a time of 833 µs, determined by the difference in the repetition rates of the two combs. The spectral resolution reached 108.4 MHz with the comb teeth-resolved structures, which is proficient for Doppler-limited spectral measurement and beneficial for revealing the interaction between the fine structure and magnetic field of paramagnetic molecules. This technology has been further applied to the measurement of the optical activity (as weak as $10^{-5}$) of chiral limonene in liquid phase with a sampling time of 5.4 s.

## Results
### Principle and experimental setup

Figure 1a illustrates the basic principles of the cross-polarization experiment based on DCS. The paramagnetic sample in the longitudinal magnetic field is excited by a broadband linearly polarized frequency comb. The electric fields of the signal comb (SC) and the local oscillator (LO) comb are given by

$$\widetilde{E}_{SC,\parallel}(\omega) = \sum_n A_{SC,n}\delta[\omega - (f_{SC,0} + nf_{SC,r})] \tag{1}$$

$$\widetilde{E}_{LO,\perp}(\omega) = \sum_n A_{LO,n}\delta[\omega - (f_{LO,0} + nf_{LO,r})] \tag{2}$$

where $f_{SC/LO,r}$ and $f_{SC/LO,0}$ denote the repetition rate and the optical comb's optical offset frequency of the SC (LO) comb, respectively. The unit impulse function $\delta$ is simply used to represent the frequency comb structure. $A_{SC/LO,n}$ is the complex amplitude of the $n$-th tooth, and $n$ is the comb line index. Linearly polarized light is equivalent to the superposition of left-circularly polarized (LCP) and right-circularly polarized (RCP) light, which correspond to the Zeeman subtransitions of $\triangle M_J = -1$ and $\triangle M_J = +1$, respectively. The sample's response function to the different circularly polarized light can be expressed as[38]

$$\widetilde{H}_{RCP}(\omega) = 1 + 4\pi^2 ic^{-1}\omega\chi_{RCP}(\omega)L \tag{3}$$

$$\widetilde{H}_{LCP}(\omega) = 1 + 4\pi^2 ic^{-1}\omega\chi_{LCP}(\omega)L \tag{4}$$

where $\chi_{RCP/LCP}(\omega)$ is the sample's linear susceptibility to RCP/LCP, $L$ is the optical length, $c$ is the speed of light, and $\omega$ is the optical angular frequency. In the molecular resonance frequency range, the difference in the response function leads to circular dichroism and circular birefringence. With the interaction between the light and the sample, the polarization state of the incident light changes, which results in perpendicular FID electric-field components,

$$\begin{aligned}\widetilde{E}_{FID,\perp}(\omega) &= E_{SC}(\omega)\widetilde{H}_{RCP}(\omega) - E_{SC}(\omega)\widetilde{H}_{LCP}(\omega)\\ &= \sum_n A_{SC,n}\delta[\omega - (f_{SC,0} + nf_{SC,r})] * 4\pi^2 ic^{-1}\omega\triangle\chi(\omega)L\end{aligned} \tag{5}$$

where the difference $\triangle\chi(\omega) = \chi_{RCP}(\omega)-\chi_{LCP}(\omega)$ between two linear susceptibilities is defined as the magnetic optical activity susceptibility. After multiheterodyne beating with the LO comb, the radiofrequency signal can be detected by a single detector without a moving part,

$$\widetilde{E}_{FID,\perp}(\omega)\widetilde{E}_{LO,\perp}^{*}(\omega) = \sum_n A_{SC,n}A_{LO,n}^{*}\delta[\omega - (\triangle f_0 + n\triangle f_r)] * 4\pi^2 ic^{-1}\omega\triangle\chi(\omega)L \tag{6}$$

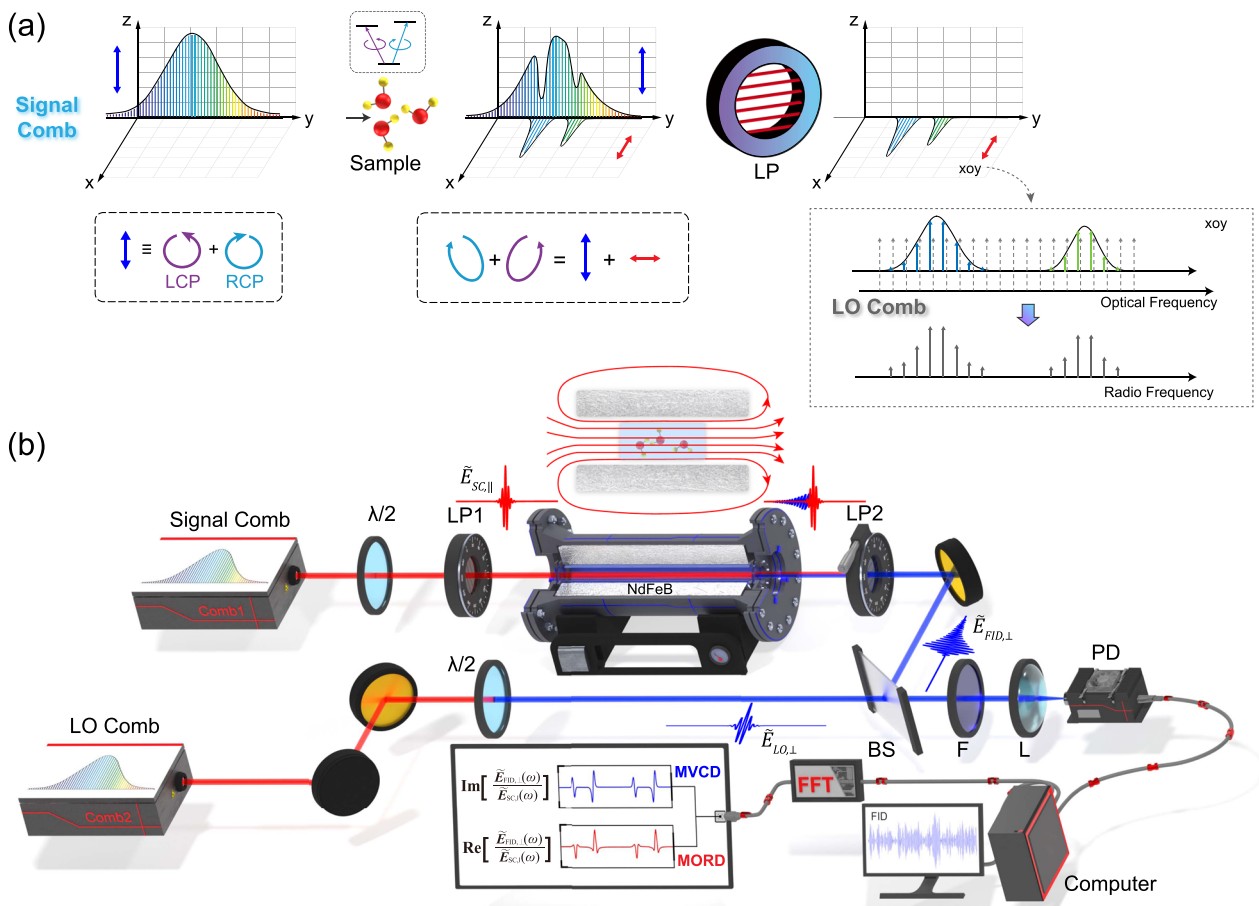

**Fig. 1 | Basic principles and experimental setup of the DC-OAS system. a** Sketch of the basic principles of the magnetic optical activity (MOA) response of the frequency comb in the cross-polarization scheme. The signal comb (SC) polarized along z axis, is incident on the sample cell in the magnetic field. For the Zeeman-split absorption lines of paramagnetic molecules, the left- and right-circularly polarized components (LCP: $\triangle M_J = -1$; RCP: $\triangle M_J = +1$) of the linearly polarized field have different wavelength-dependent complex propagation constants, which results in magnetic optical rotatory dispersion (MORD) and magnetic vibrational circular dichroism (MVCD). After interacting with the paramagnetic sample, the polarized direction of the spectral components involved in the resonant transition rotates, resulting in MOA signal polarized along x axis. The MOA signal in optical frequency is down-converted to radio frequency by multiheterodyne beating with the local oscillator (LO) comb. The linear polarizer (LP) oriented along x axis is used to filter out the non-MOA spectral components. **b** Schematic of the DC-OAS system. The ultrashort mid-infrared pulse emitted from the SC passes through a half-wave plate ($\lambda/2$) and linear polarizer LP1 to ensure linear polarization. Through interaction with paramagnetic molecules in the magnetic field of the NdFeB magnet, vertical components of the electric field are created as free-induced decay (FID). Only the MOA FID response is extracted by linear polarizer LP2, and is combined with the beam from the LO comb by a beam splitter (BS). The repetition rates between two combs are slightly different to obtain a dual-comb asynchronous optical sampling of the electric response of MOA FID. After an optical filter (F) to avoid aliasing, the interferogram signals are focused on the photodetector (PD) by a lens (L) and converted into electrical signals, which are recorded digitally by a data-acquisition card (Alazar Tech, ATS9350).

where $\triangle f_0 = f_{SC,0} - f_{LO,0}$ and $\triangle f_r = f_{SC,r} - f_{LO,r}$. The imaginary part of the magnetic optical activity susceptibility $\triangle\chi(\omega)$ corresponds to the MVCD spectrum, and the real part corresponds to the MORD spectrum[39]. Therefore, these components include almost all the information on the magnetic optical activity. In this scheme, only the optical frequencies at which the MOA occurs are acquired in discrete comb modes. The above principle analysis suggests the two characteristics of DC-OAS: first, the selective detection of MOA response avoids artefacts and correction processes, and eliminates the interference of non-MOA molecular response; second, the signal-to-noise ratio (SNR) and frequency precision of MOA spectroscopy are greatly improved, benefitting from the excellent characteristics of OFC[28,38], which will increase the application of MOA measurements in precise spectroscopy.

Figure 1b depicts the experimental configuration of our DC-OAS system. The home-made mid-infrared dual-comb spectrometer, has been introduced in our previous works based on the optical modulation technique[40,41], the repetition frequency of which is 108.4 MHz and

the difference is 1.2 kHz. When a mid-infrared ICL centered at 3.37 μm is served as the signal of the nonlinear optical modulation process, the power of our dual-comb sources are 150 mW and 190 mW, respectively, and the spectral coverage is 3.30–3.60 μm. After the linear polarizer LP1, the signal comb (SC) propagates through a 7.6-cm-long gas cell located inside a cylindrical hollow NdFeB magnet (see Supplementary Information note 3 for magnetic field distribution). The MOA-FID electric field is extracted by LP2 oriented perpendicularly and combined with the LO comb to obtain the interferogram $S_{FID,\perp}(t)$, which can be expressed as the complex function $\widetilde{E}_{FID,\perp}(\omega)\widetilde{E}_{LO,\perp}(\omega)$ in the frequency domain. The signal is acquired by a mid-infrared photodetector (VIGO PVI-4TE-3.4). To eliminate the influence of the background and non-MOA signal, the extinction ratio of linear polarizers (Meadowlark Optics) is approximately $10^{-6}$ @3.4 μm. In order to restore the MVCD and MORD spectra of gaseous molecules, the complex electric field $\widetilde{E}_{SC,\parallel}(\omega)$ must be measured independently. We evacuated the sample cell and rotated LP2 by a small angle of α = 0.45°. Thus, the interferogram signal $S_{SC,\parallel}(t)$ is recorded by the detector,

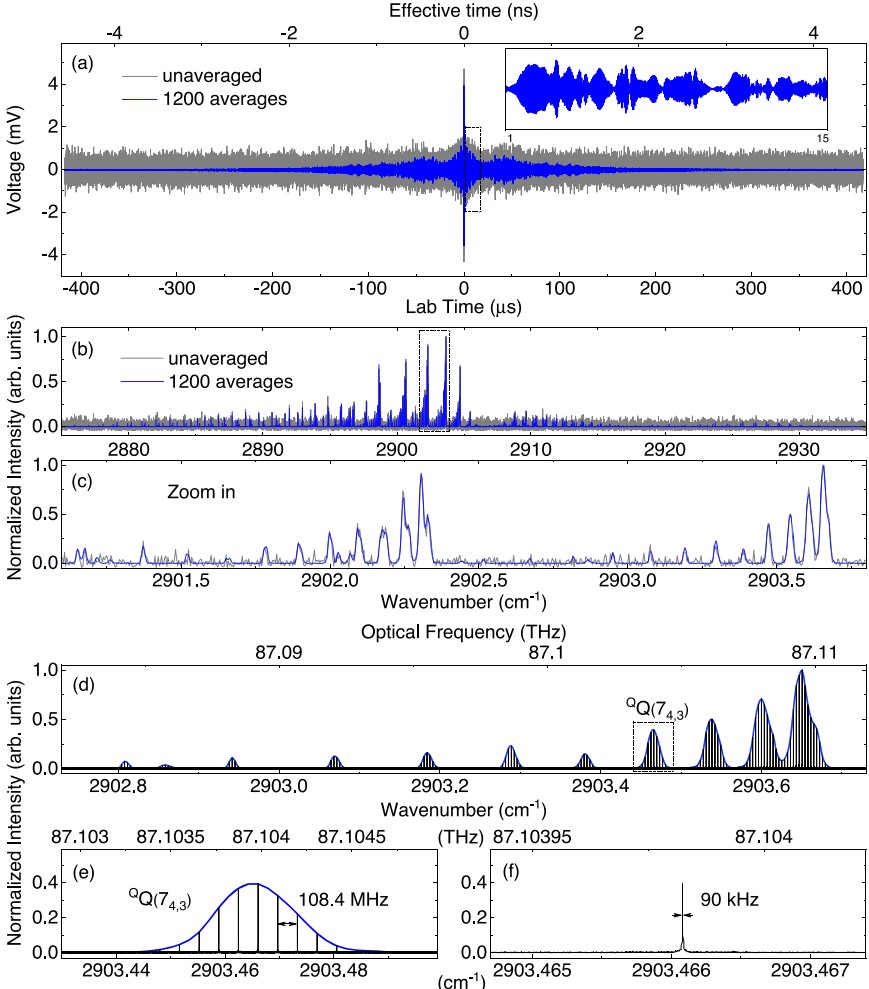

**Fig. 2 | Experimental interferograms and mode-resolved DC-OAS spectra.**
**a** Comparison between the unaveraged and 1200-average interferograms within a single period of 833 μs; the inset plot is the induced-free decay of magnetic optical activity (MOA) within 14 μs. **b** Spectra of the MOA signal for 833 μs (gray line) and 1 s (blue line) measurement times. **c** Partial spectra of **b** ranging from 2901 cm⁻¹ to 2904 cm⁻¹ of the MOA. **d** Mode-resolved MOA spectrum of nitrogen dioxide with a recording time of 1 s. The spectral range from 87.08 THz to 87.10 THz is shown to reveal the series lines of the $^{Q}Q(K_a = 4)$ subbranch; the blue curve is the contour fitting. **e** Line of the $^{Q}Q(7_{4,3})$ transition is composed of several combs with a 108.4 MHz spacing. **f** the single comb with 90 kHz linewidth.

whose Fourier transformation corresponds to the complex function $\widetilde{E}_{SC,\parallel}(\omega)\widetilde{E}_{LO,\perp}(\omega)\sin\alpha$. Therefore, the magnetic optical activity susceptibility can be directly obtained by

$$\triangle\chi(\omega) \propto \frac{\widetilde{E}_{FID,\perp}(\omega)}{\widetilde{E}_{SC,\parallel}(\omega)} = \frac{\widetilde{E}_{FID,\perp}(\omega)\widetilde{E}_{LO,\perp}^{*}(\omega)}{\widetilde{E}_{SC,\parallel}(\omega)\widetilde{E}_{LO,\perp}^{*}(\omega)} = \frac{\mathfrak{F}\left[S_{FID,\perp}(t)\right]}{\mathfrak{F}\left[S_{SC,\parallel}(t)\right]/\sin\alpha} \quad (7)$$

where the operator $\mathfrak{F}$ refers to Fourier transform.

## Performance characterization of DC-OAS

Attracted by the paramagnetic properties due to unpaired electrons, we choose a triatomic molecule (nitrogen dioxide, $NO_2$), one of the naturally stable free radicals, as the sample to demonstrate the performance of DC-OAS. The sample cell was filled with gaseous $NO_2$ at a concentration of 20% in nitrogen ($N_2$) buffer gas at a total pressure of 40 mbar. Figure 2a compares the unaveraged and 1200-average detector signals within an interferometric period of 833 μs, corresponding to the inverse of the repetition frequency difference, $1/\triangle f_r$. The unaveraged interferogram depicts the time domain response of the MOA, although the signal level is comparable to the detector noise. Benefitting from the long-term coherent averaging of DC-OAS, more details in the time domain can be retrieved from the detector noise.

As the blue line shown in Fig. 2a, an interferogram with a clear time-domain FID (inset in Fig. 2a) was obtained after 1200 averages, which corresponds to a measurement time of 1 s. The centerburst observed in Fig. 2a includes the residual background spectrum in the crossed-polarization scheme, which was attributed to the imperfection of polarizers and anisotropy of cell windows. Due to the weak energy, it had the negligible contribution in our experiment. Figure 2b compares the DC-OAS spectra after Fourier-transformation for the unaveraged interferogram (833-μs acquisition time) and the 1200-average interferogram (1-s acquisition time) without extra data processes. The portion from 2901 to 2904 cm⁻¹ is shown in Fig. 2c. The peak signal-noise-ratio (SNR) @2903.65 cm⁻¹ reach to be 29.60 for the measurement time of 833 μs, which is competent to reveal the significant spectral components of MOA although the time-domain response signal is masked by the detector noise. The results demonstrate the ability of our technique to achieve the high-sensitivity and rapid measurement.

The intensity spectrum of DC-OAS with unique mode-resolved structure was obtained after Fourier transforming the raw data stream of the time length of 1 s. The mode-resolved structure of the $^{Q}Q(K_a = 4)$ subbranch is shown in Fig. 2d–f. The magnified view of the $^{Q}Q(7_{4,3})$ transition reveals a comb-mode structure of MOA lines with a 108.4 MHz spacing and a 90-kHz linewidth. At a measurement time of 1 s, the peak SNR is up to 1020 at approximately 2903.65 cm⁻¹

corresponding to the transition of $^{Q}Q(4_{4,1})$. The average SNR of the full spectral range from 2850 to 2950 cm$^{-1}$ was calculated to be 443.90. Across the full spectral coverage of DCS, the figure of merit ($SNR \times M/T^{\frac{1}{2}}$) of our scheme is ~$1.04 \times 10^{7}$ Hz$^{1/2}$, where $M$ ($2.3 \times 10^{4}$) is the number of comb teeth within the coverage of the dual-comb system and $T$ (1 s) is the measurement time. In the experiment with 50 mW of input optical power of the Signal Comb, a MOA signal power of approximately 0.5 µW (derived from the voltage of ~8 mV and the responsivity of 16 mV µW$^{-1}$) is injected into the photodetector. The power of pure OA signal analyzed by the polarizer is very weak so that the resulting noise is much lower than the total noise. If multiwatt mid-infrared sources[42,43] are introduced in this experiment, the SNR can be improved to a certain extent.

## Intensity spectrum of MOA

The intensity spectrum of MOA, also defined as the magnetic rotation spectrum in the Methods section[44,45], is an intuitive manifestation of the MOA response of molecular transitions for the combined action of magnetic vibrational circular dichroism and magnetic optical rotatory dispersion[46]. As shown in Fig. 3a, the spectral span covers the $v_1+v_3$ cold band and the $v_1+v_2+v_3-v_2$ hot band of NO$_2$ from 2870 to 2940 cm$^{-1}$. The line strength of the $v_1+v_2+v_3-v_2$ hot band is approximately 50 times weaker than that of the $v_1+v_3$ band. The peak SNR culminates at 32100 at approximately 2903.65 cm$^{-1}$ for a measurement time of 1000 s.

The lines in the Q-branch are stronger than those in P- and R-branches. This is attributed to the fact that the line intensity assignments of the Zeeman subtransitions in the Q-branch are different from those of the P(R)-branch lines[44]. As indicated by the subbranches of $^{Q}P(K_a=4)$, $^{Q}Q(K_a=4)$ and $^{Q}R(K_a=4)$ shown in Fig. 3b–d, our system demonstrates superior capabilities (broadband and high sensitivity) to attain the observation of three branches with $K_a=1$–10, $N=K_a$-21, where $N$ is the angular momentum due to the rotation of the nuclear framework and $K_a$ is the projection of the rotational angular momentum onto the inertial axis of smallest moment. For the NO$_2$ molecule, the spin-rotation splitting $\triangle v_{sr}$ varies with the quantum numbers $N$ and $K_a$. Due to broadened linewidth caused by collisions and by the Doppler effect, ~350-MHz linewidth in the experiment, only doublets with low $N$ and high $K_a$ can be resolved. The same characteristics of MOA intensity spectrum are observed in Fig. 3e, c. The lines with $N=8$–21 of the $^{Q}Q(K_a=8)$ subbranch are resolved doublets of spin-rotation splitting, while $^{Q}Q(K_a=4)$ subbranch is unresolved, as shown in Fig. 3e, c.

In addition, the weak lines of the $v_1+v_2+v_3-v_2$ hot band are depicted in Fig. 3f. The plot (y-scale expanded 1000-fold) shows the measured $^{Q}Q(K_a=4)$ subbranch and $^{Q}Q(K_a=3)$ subbranch of this hot band, although the spectral line strength is so weak (50 times weaker than the $v_1+v_3$ band). The technique with high resolution and high sensitivity directly reveals the MOA intensity of multiple spectral lines, which simplifies the spectral line frequency analysis for paramagnetic molecules.

## Magnetic vibrational CD and ORD of nitrogen dioxide

The VCD and ORD spectra of the MOA response can be simultaneously retrieved from the measured and reference interferograms of the DC-OAS technique. More details can be seen in the section of Principle and Experimental Setup. From Eq. (4), the MVCD ($\triangle$A) and MORD ($\triangle\varphi$) spectra in a magnetic field can be directly expressed as[18,38]

$$\triangle A = \frac{4}{2.303} \text{Im}\left(\frac{\tilde{E}_{FID,\perp}(\omega)}{\tilde{E}_{SC,\parallel}(\omega)}\right) = \frac{4}{2.303} \text{Im}\left(\frac{\mathfrak{F}[S_{FID,\perp}(t)]}{\mathfrak{F}[S_{SC,\parallel}(t)]/\sin\alpha}\right) \quad (8)$$

$$\triangle\varphi = \text{Re}\left(\frac{\tilde{E}_{FID,\perp}(\omega)}{\tilde{E}_{SC,\parallel}(\omega)}\right) = \text{Re}\left(\frac{\mathfrak{F}[S_{FID,\perp}(t)]}{\mathfrak{F}[S_{SC,\parallel}(t)]/\sin\alpha}\right) \quad (9)$$

which are the embodiment of magnetic optical activity in absorption and dispersion, respectively, and can be transformed to each other through the integration of the Kramers–Kronig relation. VCD (ORD) provides information on the whole of a molecule and on the stereochemical environment of optically active chromophoric groups[47]. Here, high-resolution and high-sensitivity VCD (ORD) spectra provide more information on gaseous NO$_2$ molecule to study the fine structure of molecules and improve the molecular model, such as transition levels, spin-rotation coupling and Zeeman behavior.

The absolute MVCD and MORD values are measured with an average time of 1000 s, as shown in Fig. 4a, b. The strongest line of the $^{Q}Q(4_{4,1})$ transition is measured with $|\triangle A|=0.25$ ($|\triangle\varphi|=0.12$ rad). The standard deviation of the noise fluctuations is considered as the 1-σ sensitivity[40], where the 1-σ sensitivity of $\triangle A$ is calculated to be $1.31 \times 10^{-4}$, and the 1-σ sensitivity of $\triangle\varphi$ is $0.60 \times 10^{-4}$ rad in a measurement time of 1000 s. From the Fig. S1 in the Supplementary Information note 1, one can see that the sensitivity of OA spectra is linear to $t^{-1/2}$, even up to the measurement time $t > 1000$ s. Figure 4c, d show the series lines of the $^{Q}Q(K_a=4)$ subbranch in the $v_1+v_3$ band. Based on spectral parameters from the HITRAN database[48] and the Zeeman behavior of molecules (see Supplementary Information note 2 for details), the simulation was completed to theoretically predict this subbranch, shown as the blue and red dotted lines. The high consistency between the measured and predicted spectra indicated the capacity of our system to measure MVCD and MORD.

The high-sensitivity, high-resolution MVCD allows us to clearly observe the lineshape characteristics of the doublets (due to the fine structure of spin-rotation coupling), which cannot be resolved in the intensity spectrum. The MVCD (MORD) lines of the $^{Q}P(14_{5,9})$, $^{Q}P(30_{0,30})$ and $^{Q}P(34_{6,28})$ transitions are depicted in Fig. 4e, g, i (4f, 4h and 4j). The frequency positions of the two different unresolved doublet transitions $N'_{Ka',Kc',S'=-1/2} \leftarrow N''_{Ka'',Kc'',S''=-1/2}$ (dotted lines) and $N'_{Ka',Kc',S'=+1/2} \leftarrow N''_{Ka'',Kc'',S''=+1/2}$ (solid lines) are marked with magenta lines in Fig. 4e–j. Due to the difference in the relative frequency positions between the spin-splitting doublets, the $^{Q}P(14_{5,9})$ transition is presented with the "V-type" lineshape, while the $^{Q}P(30_{0,30})$ and $^{Q}P(34_{6,28})$ transitions are presented with the "Λ-type" lineshape. The simulation results agree well with our measured molecular characteristics, in which the fine structures of spin-rotation coupling are masked by the linewidth broadening of the Doppler effect. Our MVCD (MORD) technique provides the potential to access fine-structure information, which is difficult to be determined by traditional spectroscopic methods.

## Magnetic vibrational CD and ORD of nitric oxide

We also measured the MOAS spectra of nitric oxide with a dual-comb spectrometer working at 2.7 µm. Here, the central wavelength of the dual-comb spectrometer was adjusted to 2.7 µm for the overtone band ($2v_0$) of NO, and the sample cell was filled with pure NO gas at 65 mbar. The spectral region ranged from 3695 to 3775 cm$^{-1}$, which covered the R-branch, Q-branch, and part of the P-branch. Figure 5a, b compare the absorption spectrum with the MOA intensity spectrum. The crossed polarizer of the dual-comb optical activity spectrometer was rotated by 0.45° to record the absorption spectrum of NO after removing the permanent magnet. Due to atmospheric water vapor in the open optical path, many absorption lines of water vapor are overlapped with portions of absorption lines of NO (red curve). Based on the selective detection of DC-OAS for paramagnetic molecules, the MOA intensity spectrum eliminates the interference from diamagnetic molecules (here is the atmospheric water vapor). This characteristic of DC-OAS provides a powerful method for the spectral calibration and analysis of unknown free radicals and molecular dynamics study of free radicals in complex reaction processes.

The MVCD and MORD spectra of NO are shown in Fig. 5c, d, respectively. The gray shaded region represents the spectral energy

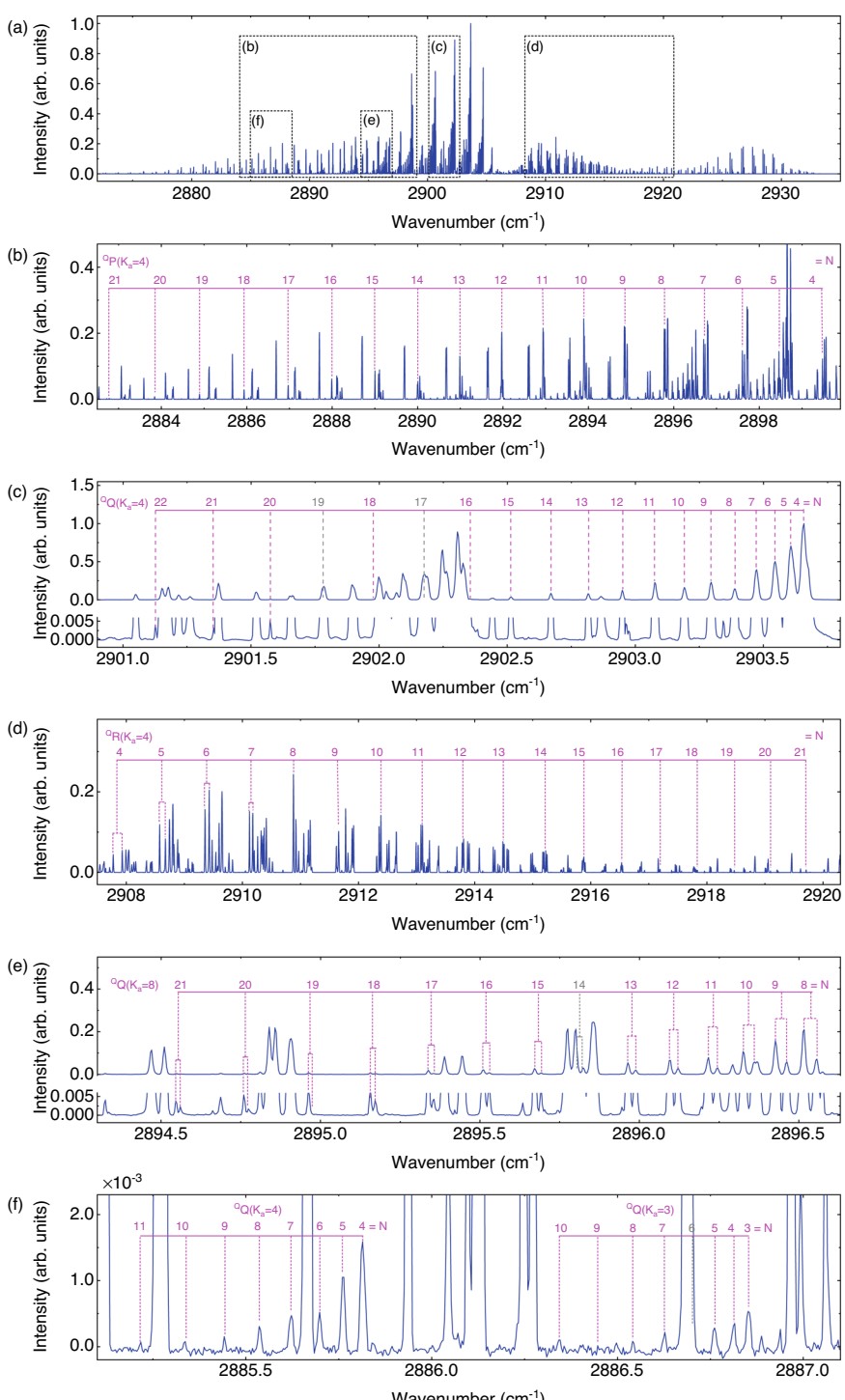

**Fig. 3 | Measured intensity spectrum of DC-OAS. a** Experimental DC-OAS intensity spectrum of $NO_2$ obtained from a continuous 1000 s recording and sampled at exactly the comb-line spacing of 108 MHz from 2870 to 2940 $cm^{-1}$. The spectrum is normalized by a background spectrum, removing the influence of spectral shape on the relative intensity of spectral lines. **b**–**f** Magnified views surrounded by rectangles in **a**. **b**–**d** Series lines from the $^QP(K_a=4)$, $^QQ(K_a=4)$, $^QR(K_a=4)$ sub-branch in the $\nu_1 + \nu_3$ band with $N = K_a$-21. **e** Series lines of the $^QQ(K_a=8)$ subbranch in cold $\nu_1 + \nu_3$ band (top) and the magnified weak spectral lines (bottom). **f** Measured intensity spectrum of DC-OAS in the $\nu_1 + \nu_2 + \nu_3 - \nu_2$ hot band.

depleted by saturated absorption due to $H_2O$ in the open optical path. Figure 5e (5f) includes the MVCD (MORD) lines of the $^2\Pi_{3/2}$ P(5.5) and $^2\Pi_{1/2}$ P(5.5) transitions, and Fig. 5i (5j) shows the MVCD (MORD) lines of $^2\Pi_{3/2}$ P(6.5) and $^2\Pi_{1/2}$ R(6.5). These lines from the $X^2\Pi_{1/2}$ and $X^2\Pi_{3/2}$ subsystems have opposite phases, which is attributed to the opposite sign of the Landé factor $g_J$[22,49]. A portion of the Q-branch is shown in Fig. 5e, h. Based on the MOA simulation of NO (see Supplementary

Information note 2 for more details), the calculated spectra (the dashed lines in Fig.5e–j) agree well with the measured spectra.

## Magnetic vibrational LD and LB of nitrogen dioxide
Finally, we measured another common magneto-optical effect, the Voigt effect, with the DC-OAS technique. For the Voigt effect existing in a transverse magnetic field[45], the $\sigma$ transitions ($\triangle m_J = \pm 1$) and $\pi$

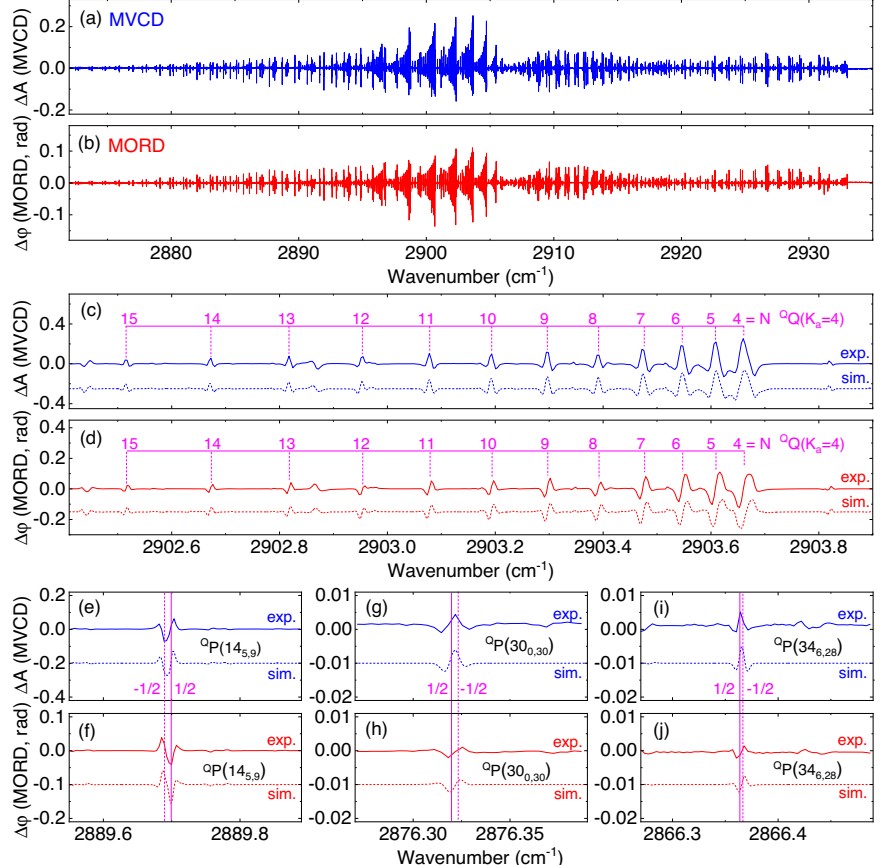

**Fig. 4 | Magnetic vibrational optical activity signals of nitrogen dioxide.**
**a, b** Retrieved MVCD and MORD spectra of $NO_2$ covered from 2870 to 2940 $cm^{-1}$, respectively. **c** and **d** MVCD and MORD spectral lines of the $^QQ(K_a = 4)$ subbranch, the dashed lines below are the simulation results. **e, g, i** (**f, h, j**) MVCD (MORD) lines of the $^QP(14_{5,9})$, $^QP(30_{0,30})$, and $^QP(34_{6,28})$ transitions. The transition frequency of the unresolved doublets (labeled −1/2 and 1/2) are marked by pink dotted and solid lines, respectively. The MVCD lineshape of the "−1/2 1/2" doublet is presented as "V-type", while the lineshape of the "1/2 −1/2" doublets are "Λ-type". The dashed lines in the above figures are the simulated results.

transitions ($\triangle m_J = 0$) are excited simultaneously by linearly polarized light perpendicular to and parallel to the magnetic field, respectively. Importantly, the analysis of the Voigt effect helps to comprehensively analyze the Zeeman behavior of molecules and complete the $\pi$ transitions not involved in the Faraday effect. Our proposed DC-OAS with a high polarization sensitivity can be used to resolve various polarization effects, including magnetic vibrational linear dichroism (LD) and linear birefringence (LB). The gas cell is filled with $NO_2$ (a pressure of 8.5 mbar and a concentration of 20%), and the effective optical path is 30 cm. The transverse magnetic field is created by two bulk NdFeB permanent magnets (5 cm×5 cm×40 cm) with a distance of 2.2 cm. The magnetic field strength is -3100 G and the nonuniformity is <3%. The sample comb is incident to excite the sample at a linear polarization of 45 degrees with the transverse magnetic field to obtain the maximum LD (LB) signal[38]. Other experimental schemes and data processing are the same as those for Faraday-configuration DC-OAS.

Figure 6a, b depict magnetic vibrational LD ($\triangle A$) and LB ($\triangle \varphi$) spectra with 200-s acquisition time, which covers the $\upsilon_1 + \upsilon_3$ band of $NO_2$ molecule. Similar to the MVCD (MORD), the strongest line is at the $^QQ(4_{4,1})$ transition with $|\triangle A| = 0.23$ ($|\triangle \varphi| = 0.12$ rad). Portions of the LD (LB) lines are shown in Fig. 6c (6d) and Fig. 6e (6f). For the $NO_2$ molecule under the transverse magnetic field, the simulations were completed by additionally considering the $\pi$ transitions (see Supplementary Information for more details). The calculated results, shown by the dashed lines in Fig. 6c–f, agree with the experimental spectra, which demonstrates the portability of this technical scheme in LD (LB) measurement. This system measured the LD and LB spectra without

any modification, which provides an alternative method for studying the orientation of biopolymers.

## Vibrational OA spectra of liquid-phase chiral limonene

In order to demonstrate the ability of our scheme with chiral optical activity measurement, we provided the proof-of-principle results on the (R)- and (S)-limonene and their racemic mixture, where these samples are dissolved by $CCl_4$ to the concentration of 110 mM. This experiment is performed without any other modification except that the repetition frequency difference is set to 600 Hz. In this case, the period of a single interferogram is about 1.67 ms ($1/\triangle f_r$), where only 2.5-μs-long raw data is sampled. Figure 7a shows the measured signal and reference interferograms, where the apodized resolution is 2.4 $cm^{-1}$. The measurement for each kind of limonenes lasts sixty minutes, which is limited by the refresh rate of the interferograms. In fact, the total sampling time of the available interferograms is only 5.4 s. The low duty cycle is caused by the relatively low repetition rate of our dual-comb source, which can be overcome by the high-repetition-rate frequency combs[31]. Figure 7b, c compare VCD and ORD spectra of the R- (blue line) and S-limonene (pink line) and their racemic mixture (red line), respectively. The baseline from the sample cell has been removed by measuring the reference spectra of the pure $CCl_4$. As expected, the VCD and ORD signals of R- and S-limonene are nearly identical but their signs opposite to each other, while the optical activity signals of the racemic mixture does not exhibit observable structure. These results demonstrate the capability of our technique to measure chiroptical activity as weak as $10^{-5}$ and characterize chiral

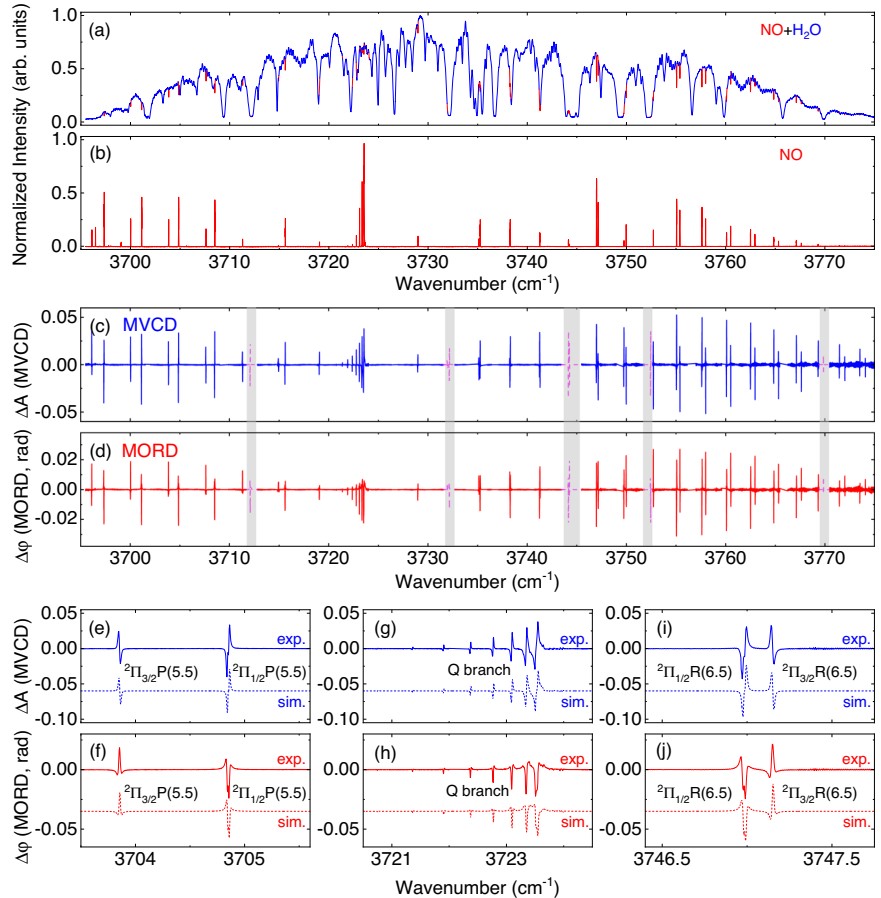

**Fig. 5 | Measured magnetic vibrational optical activity signals of nitric oxide with DC-OAS in atmospheric environment. a, b** Comparison of the absorption spectrum and MOA intensity spectrum of nitric oxide. The dips with red curves represent the absorption lines of NO. **c, d** MVCD and MORD spectra measured by DC-OAS. Gray shaded area represents the frequency area masked by $H_2O$ saturated absorption, and missing spectral lines are completed with calculated data (magenta dashed lines). The subplots of **e, g, i** (**f, h, j**) depict partial MVCD (MORD) lines from the P-branch, Q-branch, and R-branch, and the dashed lines in the above figures are the simulated results.

species. These results further demonstrate the potential of our technique to rapidly measure weak chiroptical activity signals, which is expected to advance time-resolved VCD spectroscopy to study the secondary structure dynamics of bio-chiral molecules.

## Discussion

In this study, we demonstrated that frequency combs enable high-resolution, high-sensitivity and high-speed DC-OAS, and are extended to the application of chiral species characterization. The high resolution introduced by DCS enabled magnetic optical activity spectroscopy in precision experiments, such as the analysis of molecular fine structure and the measurement of vibrational magnetic dipole moment. The sub-millisecond measurement time of DC-OAS provides a tool for the real-time monitoring of products, which is very attractive for the study of high-energy magnetic fields that cannot be operated for a long time. This technique achieved a sensitivity of $\triangle A \sim 1.31 \times 10^{-4}$@1000 s and $\triangle\varphi \sim 0.6 \times 10^{-4}$ rad @1000 s, with a 108.4-MHz resolution over a vibrational band. Our set-up has been applied in chiroptical activity measurement of liquid-phase system within the several-second sampling time, though the OA signal is $10^5$-$10^6$ times smaller than absorption. The 2.4-cm$^{-1}$ resolution (-72 GHz) via apodization in our paper is acceptable for the spectral features of the liquid/solid-phase system. However, it would be ideal if the repetition rate $f_r$ of OFC was equal to the desired resolution, and the measurement time could be reduced by more than five orders of magnitude from 1 hour to millisecond level[35]. With the development of the high-repetition-rate sources used in DCS[50,51], the millisecond-level measurement time of weak COA in liquid/solid-phase system can be achieved. In addition, based on the femtosecond characterization of OFCs, time-resolved VCD spectroscopy with femtosecond time resolution can be realized to observe ultrafast transient dynamics, such as those occurring during protein folding, asymmetric chemical reactions and free radical dynamics. The DC-OAS system proposed in this paper has advanced performance in the characterization field of optical activity measurement, and is expected to become a powerful tool for exploring the structure of chiral molecules.

## Methods

### Sensitivity of the DC-OAS

The detection limit of DC-OAS is defined as the 1-$\sigma$ sensitivity, where 1-$\sigma$ is the standard deviation of the spectral power density in the given spectral region[52]. The averaged uncertainty $\sigma$ can be expressed by[53]

$$\sigma \propto (M\sqrt{f_r/\nu_{res}})/\sqrt{T},$$

where $M$ is the number of the resolved spectral elements, $f_r$ is the repetition rate of the frequency comb, $\nu_{res}$ is the spectral resolution, and $T$ is the measurement time.

### Peak and average signal-to-noise ratio of the DC-OAS

The peak SNR is defined as the ratio of the peak signal $A_0$ to the average of the standard deviation of the spectral power density in the

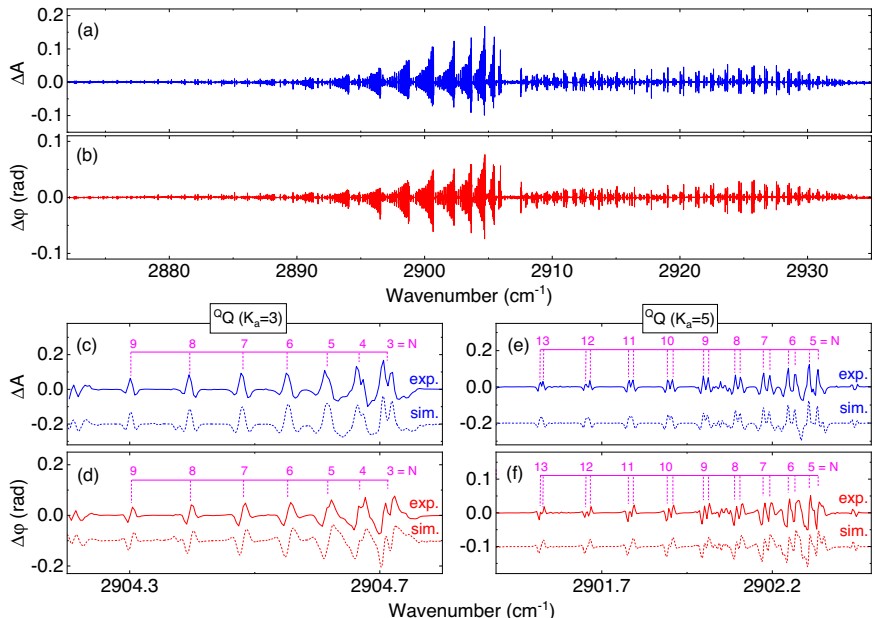

**Fig. 6 | Magnetic vibrational linear dichroism (LD) and linear birefringence (LB) of nitrogen dioxide. a** Differential absorbance $\triangle A$ between orthogonally polarized light ($\triangle A = A_\pi - A_\sigma$); **b** Differential phase $\triangle \varphi$ between orthogonally polarized light ($\triangle \varphi = \frac{\pi L}{\lambda}(n_\pi - n_\sigma)$). **c** LD (LB) (**d**) lines of the $^QQ$ ($K_a = 3$) subbranch with $N = 3-9$; **e** LD (LB) lines of the $^QQ$ ($K_a = 5$) subbranch with $N = 5-13$ (**f**). The dashed lines below is the simulated results.

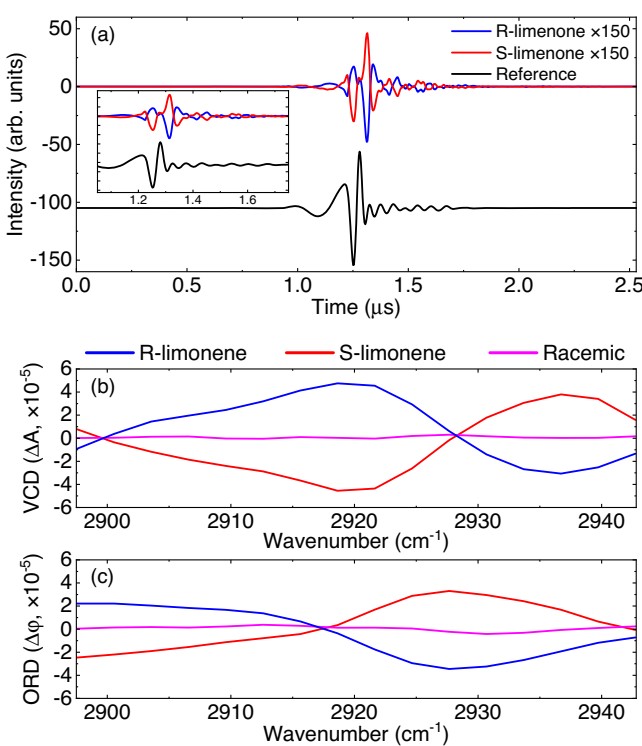

**Fig. 7 | Vibrational optical activity signal of chiral limonenes. a** The signal interferogram (black line, after averaging over 1 h) and reference interferogram (red line, after averaging over 5 min) through the R-limonene sample within 2.5 μs. The signal interferogram is magnified 150 times for comparison. The chiral limonenes dissolved in $CCl_4$ with the analyte concentration of 110 mM and the path length of 1 mm. Inset: zoom-in plot of the interferograms. **b**, **c** Absolute VCD ($\Delta A$) and ORD ($\Delta \phi$) spectra measured by DC-OAS with the samples of R-limonene (blue line) and S-limonene (red line) and their 1:1 racemic solution (pink line).

given spectral region $\sigma$,

$$SNR_{peak} = A_0/\sigma.$$

And the average SNR is calculated as the sum of the SNRs of the measured comb lines divided by the number of the measured comb lines[54].

In an idea system of dual-comb spectrometer, the SNR and the sensitivity scales nearly as $\sqrt{t}$, where $t$ is the measurement time. From Fig. S1 in Supplementary Information note 1, the performance of our DC-OAS also conforms to this characteristic, even with measurement times as long as 1000 s.

### Definition of magnetic rotation spectroscopy

The magnetic rotation spectroscopy is the result of the combined action of circular magnetic birefringence (MCB) and magnetic circular dichroism (MCD), which can be expressed by[55]

$$I_{MRS} = 1/4[T_+^{1/2} - T_-^{1/2}]^2 + (T_+ T_-)^{1/2}\sin^2\theta,$$

where $T_+$ and $T_-$ are the transmission spectra of left- and right-circularly-polarized light. And $\theta = \triangle n L \pi/\lambda$ is the Faraday rotation angle, where $\triangle n$ is the refractive index difference of left- and right-circularly-polarized light, $L$ is the distance through a medium and $\lambda$ is the light wavelength.

## Data availability

All data supporting the findings of this study are available within the article and Supplementary Information. Source data are provided as a Source Data file. Source data are provided with this paper.

## Code availability

The fast Fourier transforms (FFT) in the MATLAB program is simply used to process our dual-comb optical activity signal. And the code that simulates the Zeeman effect of nitrogen dioxide and nitric oxide

and supports magnetic-optical-activity processing in this study are available from the corresponding author upon reasonable request.

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

## Acknowledgements

The authors thank Renjun Pang for helpful discussions. W.L. and D.L. were supported in part by the National Natural Science Foundation of China, No. 12134004 & No. 12104162. W.L. was supported by the National key R&D Program of China, No. 2018YFA0306301. W.L. was supported by the Shanghai Municipal Science and Technology Major Project. W.L. was supported by the Research Funds of Happiness Flower ECNU, No. 2021ST2110.

## Author contributions

W.L. led this work. D.P. and C.G. performed the experiments and the simulations. Z.Z., Y.D., X.Z., and L.T. also discussed and analyzed the results. C.G., D.P., and W.L. prepared the paper. L.D., D.L., and Y.L. help to prepare the experimental setup and review the paper.

## Competing interests

The authors declare no competing interests.
