## [Peer review file · Nature Communications]

REVIEWER COMMENTS

Reviewer #1 (Remarks to the Author):

In this paper, the authors apply dual comb spectroscopy to measurements of magnetic optical activity. As far as I'm aware, this is the first demonstration of this application of dual comb spectroscopy and enables high spectral resolution magnetic optical activity measurements and Faraday rotation spectroscopy. Overall, it is a nice demonstrate of the technique, and the data looks very nice. I only have a few questions/comments.

1. My biggest question is how does the sensitivity compare to other techniques? The claim is that the DCS-based measurements should have high sensitivity, but there aren't any numbers for comparison. It would be good to prove this argument more.

2. Fig 1 -

a) LP and P both used for polarizers

b) It is hard to read the FFT label and the labels for the Re and Im parts

c) Component "F" not defined (assume it's a filter)

d) Colors in the inset of the comb mode sampling of xoy are hard to see. Maybe use a different color for the LO comb and also use dashed lines?

3. The quoted sensitivity on p 3 needs associated measurement time. It reads as if this sensitivity and the sub-millisecond time are both possible. This is also true in the discussion section where the sensitivity is quoted without measurement. Also, do you have an Allan deviation of the ΔA and $\Delta \phi$?

4. On page 6, how is the SNR defined? I assume that it is the peak over the measurement noise? In this case, how do you define the average SNR? Do you use the same peak signal strength and look at noise across the spectrum?

5. How much total power is on the detector? How do you know that you aren't limited by RIN instead? I don't think that the fact that the "The signal power is much lower than the detector saturation power," immediately implies that you are detector noise limited.

6. I think that the intensity MOA is also Faraday rotation, correct? It would be good to clarify that since some readers will only be familiar with one set of terms. Also, there has been a demonstration of frequency comb Faraday rotation spectroscopy (not dual comb) that would be worth citing:

Johansson, Alexandra C., Jonas Westberg, Gerard Wysocki, and Aleksandra Foltynowicz. "Optical Frequency Comb Faraday Rotation Spectroscopy." *Applied Physics B* 124, no. 5 (May 1, 2018): 79. <https://doi.org/10.1007/s00340-018-6951-8>.

7. Fig 5. How was the absorption spectrum in (a) recorded? Is it just the same way that the reference spectrum is obtained (without purging the cell)?

8. Fig 6 caption: Linear birefringence (LD) should be (LB)

Reviewer #2 (Remarks to the Author):

The authors report on the mid-infrared dual-comb optical activity spectroscopy that enables fast measurement of chiral molecules with high spectral resolution. They applied their technique to the molecular spectroscopy of NO₂ and NO to observe Zeeman-split absorption lines and compared their experimental results with theoretical calculations. The paper is well written, and the validity of the data is adequate. Because optical activity measurements of chiral molecules surely broaden the applicability of the dual-comb spectroscopy, I recommend to publish the article to Nature Communications after some minor revisions as described below.

(1) In the Introduction, the authors refer to a time-resolved VCD spectroscopy, but they do not perform the experiment in this article. This is a confusing statement, and it is better to describe it in conclusion as a future perspective.

(2) In the principle and experimental setup. A detail of the mid-infrared optical frequency combs they used is required even though they cite their previous article (Ref. 38). At least, center wavelength, spectral bandwidth, and the output power of both combs are required.

(3) Schematic figure of the experimental configuration of the NdFeB magnet is required to explain how they apply magnetic field to the gas molecule.

(4) How about the polarization stability of the signal comb? Instability of the polarization of the signal comb will cause the intensity fluctuation of the comb's power after passing through the polarizer.

(5) It seems that the denominator of the most right-hand side of Eq. (5) is not correct. It should be " $F[S_{SC}(t)] / \sin \alpha$ ", isn't it? Please also check Eqs. (6) and (7).

(6) For the theoretical calculations in the Supplementary Information (SI). The authors calculate the MVCD and MORD spectra referring Reference S11, but with little information. For example, according to S11, the Voigt-shape function is a function of self-collision HWHM and foreign gas collision HWHM, but the concrete values for their calculation are not indicated in the SI. The authors should describe the details of the functions they used for the theoretical calculations.

Reviewer #3 (Remarks to the Author):

This manuscript demonstrates the detection of magnetic field induced optical activity using a dual-comb approach. The approach is a rather straightforward modification of standard dual-comb spectroscopy, namely the LO comb has linear polarization that is orthogonal to that of the signal comb. The method is demonstrated on a gas of nitrogen dioxide that is placed in a magnetic field.

The results appear to be valid and the manuscript is generally well written, although it could use review/copy editing by a native English speaker to help scrub some minor issues with language usage, although none of them interfere with understanding the technical content.

Although the results appear correct and valid, the authors do not address the need for this advance nor the impact it may have. Generic statements are made about improving the spectral resolution and sensitivity, which are all reasonable. However, it is not clear that there are any situations where these improvements would be ground breaking and enable the discovery of new science, or be technologically relevant.

Given that it reports an incremental improvement on the detection of optical activity and similarly is a variant on standard dual comb spectroscopy, I do not see that this manuscript has the novelty or impact required for publication in Nature Communications.

Response to Reviewers

Thanks for your kind consideration and reviews on our manuscript (Manuscript ID: NCOMMS-22-27570). We reviewed the questions and suggestions carefully, which help us to improve our manuscript and strengthen the impact. According to the reviewers' comments, the manuscript has been revised with our best attention.

Note that reviewers' comments are marked in black, and authors' responses are in blue. **The corrections in the revised manuscript are marked in blue italic bold.** All changes in the manuscript text file are highlighted in blue.

Reviewer #1:

In this paper, the authors apply dual comb spectroscopy to measurements of magnetic optical activity. As far as I'm aware, this is the first demonstration of this application of dual comb spectroscopy and enables high spectral resolution magnetic optical activity measurements and Faraday rotation spectroscopy. Overall, it is a nice demonstrate of the technique, and the data looks very nice.

Reply: We appreciate the reviewer for the high evaluation of our work. The purpose of this paper is to demonstrate the potential of dual-comb spectrometer in the field of optical activity spectroscopy measurement, and we will continue to develop the application based on this technology.

I only have a few questions/comments.

1. My biggest question is how does the sensitivity compare to other techniques? The claim is that the DCS-based measurements should have high sensitivity, but there aren't any numbers for comparison. It would be good to prove this argument more.

Reply:

1. We define the sensitivity of the system in the **Methods** section of the revised manuscript.

2. We compare several representative works in the table below to show the advantages of our technique, through a literature survey of work in this field.

	Measurement time	Resolution	Sensitivity
Ref. 1	>8000 s(g)	3 GHz	$\sim 10^{-4}$
Ref. 2	3500 s (g)	250 MHz	$\sim 10^{-3}$
Ref. 3	Tens of minutes (l)	90 GHz	$> 10^{-5}$
Our work	1000 s (g)	108 MHz	$\sim 10^{-4}$
	5.4 s (l) [#]	72 GHz	$> 10^{-5}$

g: gas-phase system l: liquid-phase system

[#] *The total sampling time lasts only 5.4 s within 1-hour measurement time.*

In the gas-phase OA analysis, Timothy A. Keiderling et al.¹ have investigated the magnetic VCD spectra of a large number of gas-phase paramagnetic/ferromagnetic molecules using a Fourier transform spectrometer, achieving a sensitivity of $\sim 10^{-4}$ (total measurement time exceeds 8000 s) and a resolution of 3 GHz. Aleksandra

Foltynowicz's group² attempted to apply the Fourier-transform-spectrometer-based optical frequency comb to Faraday rotation spectroscopy, obtaining a spectral resolution of 250 MHz and a sensitivity of $\sim 10^{-3}$ @3500 s at the expense of long-range scanning. In contrast, DC-OAS is based on optical asynchronous sampling without any mechanical scanning, and the single spectral refresh time can be reduced to the order of microseconds. Moreover, the frequency-domain multi-heterodyne interference of dual-comb spectrometer down-converts the spectral information to the radio frequency, which suppresses the white noise⁴ ($\propto 1/f$) and greatly improves the detection sensitivity. Our scheme has simultaneously measured VCD and ORD spectra with a resolution of 108 MHz and a sensitivity of $\sim 10^{-4}$ @1000 s, which offers significant improvements in gas-phase OA analysis.

In the liquid-phase OA analysis, Minhaeng Cho's group provides orders of magnitude improvement in the measurement speed of weak chiroptical activity (COA) signals. They claimed that their technique, a heterodyned spectral interferometry based on the cross-polarization detection, realized the COA measurement of limonene (10^{-5} smaller than absorption) with a resolution of 90 GHz and reduced the measurement time from multiple hours to tens of minutes. In our experiment shown in the **Vibrational optical activity spectra of chiral limonenes in liquid-phase system** section of the revised manuscript, the VCD and ORD spectra of limonene were simultaneously achieved in the 5.4-s sampling time with 72-GHz resolution. However, limited by the low duty cycle of the low-repetition-rate system, the total measurement time lasted 1 hour. It would be ideal if the repetition rate f_r of optical frequency comb was equal to the desired resolution, and the measurement time could be reduced by more than five orders of magnitude from 1 hour to millisecond level. With the development of the high-repetition-rate sources, such as microcavity⁵ and quantum cascade laser⁶, our technique has unparalleled advantages in the rapid measurement of optical activity.

[1] Tam, C. N. & Keiderling T. A. Direct measurement of the rotational g-value in the ground vibrational state of acetylene by magnetic vibrational circular dichroism. *Chem. Phys. Lett.* **243**, 55-58 (1995).

[2] Johansson, A. C., Westberg, J., Wysocki, G. & Foltynowicz, A. Optical frequency comb Faraday rotation spectroscopy. *Appl. Phys. B* **124**, 1-8 (2018).

[3] Rhee, H. et al. Femtosecond characterization of vibrational optical activity of chiral molecules. *Nature* **458**, 310-313 (2009).

[4] Newbury, N. R., Coddington, I. & Swann, W. Sensitivity of coherent dual-comb spectroscopy. *Opt. Express* **18**, 7929-7945, (2010).

[5] Yu, M., Okawachi, Y., Griffith, A. G., Picqué, N., Lipson, M. & Gaeta, A. L. Silicon-chip-based mid-infrared dual-comb spectroscopy. *Nat. Comm.* **9**, 1-6 (2018).

[6] Hillbrand, J., Andrews, A. M., Detz, H., Strasser, G. & Schwarz, B. Coherent injection locking of quantum cascade laser frequency combs. *Nat. Photon.* **13**, 101-104 (2019).

Correction1: The discussion above has now been included in Paragraph 2 in the

Introduction section:

“Timothy A. Keiderling et al.^{21,22} utilized broadband FT spectroscopy to study the magnetic VCD of gas-phase paramagnetic/diamagnetic molecules with a sensitivity of 10^{-4} (over an hour of the scanning time), and a resolution of 0.1 cm^{-1} . Recently, Aleksandra Foltynowicz’s group introduced the optical frequency comb into Faraday rotation spectroscopy for the first time, and measured the high-resolution gas-phase spectrum using a long-range scanning Fourier spectrometer²³. The sensitivity is limited to be $\sim 10^{-3}$ @3500 s by the scanning speed and intrinsic noise of the Fourier spectrometer.”

Correction2: The **Vibrational optical activity spectra of chiral limonenes in liquid-phase system** section is added to introduce the preliminary work in liquid-phase chiroptical activity.

Correction3: The comparison and discussion of system performance has been added in the **Supplementary Information note 1**.

2. Fig 1 -

- LP and P both used for polarizers
- It is hard to read the FFT label and the labels for the Re and Im parts
- Component "F" not defined (assume it's a filter)
- Colors in the inset of the comb mode sampling of xoy are hard to see. Maybe use a different color for the LO comb and also use dashed lines?

Reply: We are grateful for reviewer’s the careful reading. In the revised manuscript, we have made the following modifications in the Fig. 1:

- The abbreviation of the line polarizer has been uniformed to be “LP”;
- The labels for FFT, Re and Im parts have been magnified to be more eye-catching;
- Component “F” denotes a filter, which is now clearly annotated in the revised caption;
- Thanks for your suggestion, Fig. 1b has been revised accordingly.

Correction1: The modifications are shown in the Fig. R1:

Fig. R1 Basic principles and experimental setup of the DC-OAS system.

Correction2: The sentence is added in the caption of Figure 1 to illustrate the component “F”:

“After an optical filter (F) to avoid aliasing, the interferogram signals are focused on the photodetector (PD) by a lens (L) and converted into electrical signals, which are recorded digitally by a data-acquisition card (Alazar Tech, ATS9350).”

3. The quoted sensitivity on p 3 needs associated measurement time. It reads as if this sensitivity and the sub-millisecond time are both possible. This is also true in the discussion section where the sensitivity is quoted without measurement. Also, do you have an Allan deviation of the Delta A and Delta phi?

Reply: The 1- σ sensitivity of DC-OAS is defined as the standard deviation of the spectral power density in the given spectral region⁸. The averaged uncertainty σ can be expressed by⁵

$$\sigma \propto (M\sqrt{f_r/v_{res}})/\sqrt{T},$$

where M is the number of the resolved spectral elements, f_r is the repetition rate of the frequency comb, v_{res} is the spectral resolution, and T is the measurement time. Therefore, the detection sensitivity of DC-OAS scales as the $1/\sqrt{t}$. The sub-millisecond time mentioned in our article is the time of single spectral measurement, while the 1- σ sensitivity of ΔA is calculated to be 1.31×10^{-4} , and the 1- σ sensitivity of $\Delta\phi$ is 0.60×10^{-4} rad in a measurement time of 1000 s. To avoid misleading, we modify the sensitivity “ $\Delta A \sim 1.31 \times 10^{-4}$ ($\Delta\phi \sim 0.6 \times 10^{-4}$ rad)” to “ $\Delta A \sim 1.31 \times 10^{-4}$ @1000 s ($\Delta\phi \sim 0.6 \times 10^{-4}$ rad @1000 s)”. The definition has been included in the **Methods** section of the revised manuscript. In addition, we measure the chiroptical activity spectra of limonene, (signal strength is as low as 10^{-5}). The results are elaborated in the **Vibrational optical activity spectra of chiral limonenes in liquid-phase system.** section of the revised manuscript, which indicates that our technique is robust and reliable.

Figure R2a and R2b show that the peak SNRs of MVCD and MORD both increase linearly with the square root of the measurement time. The Allan deviations of VCD (ΔA) and ORD ($\Delta\phi$) are shown in Fig. R2c and R2d, respectively. The linear fitting curves of VCD and ORD have the same slope of -1/2, which confirms $t^{-1/2}$ dependence characteristic of sensitivity for the measurement time (t) up to 1000 s. With an integration time of 1000 s, the detection sensitivity of VCD and ORD reached 1.31×10^{-4} and 0.6×10^{-4} rad, respectively. This part of the discussion is added in a Supplementary Information Note 1.

Fig. R2 **a** and **b** Signal-to-noise ratio (SNR) of VCD and ORD as a function of averaging time, respectively. The solid lines are the fitting curves of the $t^{1/2}$ trends. **c** and **d** Allan deviations of the 1- σ sensitivities of VCD and ORD. The curves show the inverse square root dependency. The solid lines are the fitting curves of the $t^{-1/2}$ trends.

[8] Muraviev, A. V., Smolski, V. O., Loparo, Z. E. & Vodopyanov, K. L. Massively parallel sensing of trace molecules and their isotopologues with broadband subharmonic mid-infrared frequency combs. *Nat. Photon.* **12**, 209-214, (2018).

Correction1: The definition of the sensitivity is added in the **Methods** section:

“Sensitivity of the DC-OAS. The detection limit of DC-OAS is defined as the 1- σ sensitivity, where 1- σ is the standard deviation of the spectral power density in the given spectral region⁵³. The averaged uncertainty σ can be expressed by⁵⁴

$$\sigma \propto (M\sqrt{f_r/v_{res}})/\sqrt{T},$$

where M is the number of the resolved spectral elements, f_r is the repetition rate of the frequency comb, v_{res} is the spectral resolution, and T is the measurement time.”

Correction2: The OA sensitivity in the revised manuscript is corrected as “ $\Delta A \sim 1.31 \times 10^{-4}$ @1000 s ($\Delta\varphi \sim 0.6 \times 10^{-4}$ rad @1000 s)”.

Correction3: The sentences are added in the **Magnetic vibrational CD and ORD of nitrogen dioxide** section to explain the association between sensitivity and measurement time:

“The standard deviation of the noise fluctuations is considered as the 1- σ sensitivity⁴⁰, where the 1- σ sensitivity of ΔA is calculated to be 1.31×10^{-4} , and the 1- σ sensitivity of $\Delta\varphi$ is 0.60×10^{-4} rad in a measurement time of 1000 s. From the Fig. S1 in the Supplementary Information, one can see that the sensitivity of OA spectra is linear to $t^{-1/2}$, even up to the measurement time $t > 1000$ s.”

Correction4: **Supplementary Information note 1** is added to explain the Allan deviations of the ΔA and $\Delta\varphi$.

4. On page 6, how is the SNR defined? I assume that it is the peak over the measurement noise? In this case, how do you define the average SNR? Do you use the same peak signal strength and look at noise across the spectrum?

Reply: The peak signal-to-noise ratio (SNR) is defined as the ratio of the peak signal A_0 to the average of the standard deviation of the spectral power density in the given spectral region σ ,

$$SNR_{peak} = A_0/\sigma,$$

and the average SNR is calculated as the sum of the SNRs of the measured comb lines divided by the number of the measured comb lines^{9,10}. Therefore, the peak SNR is calculated as the peak value of the retrieved ${}^{\text{Q}}\text{Q}(4_{4,1})$ line divided by the standard deviation of the noise in the full spectral range from 2850 to 2950 cm^{-1} . The average SNR of DC-OAS is obtained by averaging the SNRs of all measured signals¹¹. The definitions of the peak SNR and the average SNR are provided in the **Methods** section of the revised manuscript, which is much more convenient for readers to understand.

[9] Tomberg, T., Muraviev, A., Ru, Q. & Vodopyanov, K. L. Background-free broadband absorption spectroscopy based on interferometric suppression with a sign-inverted waveform. *Optica* **6**, 147-151 (2019).

[10] Wang, Q. et al. Dual-comb photothermal spectroscopy. *Nat. Commun.* **13**, 2181 (2022).

[11] Yan, M., Luo, P. L., Iwakuni, K., Millot, G., Hänsch, T. W. & Picqué, N. Mid-infrared dual-comb spectroscopy with electro-optic modulators. *Light Sci. Appl.* **6**, 17076-17076, (2017).

Correction: The definition of the signal-to-noise ratio is added in the **Methods** section: **“Peak and average signal-to-noise ratio (SNR) of the DC-OAS. The peak SNR is defined as the ratio of the peak signal A_0 to the average of the standard deviation of the spectral power density in the given spectral region σ ,**

$$SNR_{peak} = A_0/\sigma.$$

And the average SNR is calculated as the sum of the SNRs of the measured comb lines divided by the number of the measured comb lines⁵⁵.”

5. How much total power is on the detector? How do you know that you aren't limited by RIN instead? I don't think that the fact that the "The signal power is much lower than the detector saturation power," immediately implies that you are detector noise limited.

Reply: In the section of “Performance characterization of DC-OAS”, “*In the experiment with 50 mW of input optical power, a MOA signal power of approximately 0.5 μW (derived from the voltage of ~ 8 mV and the responsivity of 16 mV μW^{-1}) is injected into the photodetector*”, the power of signal comb on the detector is ~ 0.5 μW and the power of Local Oscillator (LO) comb is attenuated to ~ 30 μW to avoid detector nonlinearities. Our intention is to show that the power of signal comb is very low and the resulting

noise is much lower than the total noise. In this case, boosting the signal comb's power can increase the SNR. We agree with the reviewer's conception and revise the manuscript as follows:

Correction: we have replaced "*The signal power is much lower than the detector saturation power, so the SNR is mainly limited by the noise of the detector*³⁹⁻⁴¹. Thus, if multiwatt mid-infrared sources^{42,43} are introduced in the experiment, the SNR will be improved furtherly by 1-2 orders, which is attractive for higher-sensitivity measurements of OA" by "*The power of pure OA signal analyzed by the polarizer is very weak so that the resulting noise is much lower than the total noise. If multiwatt mid-infrared sources^{42,43} are introduced in this experiment, the SNR can be improved to a certain extent*".

6. I think that the intensity MOA is also Faraday rotation, correct? It would be good to clarify that since some readers will only be familiar with one set of terms. Also, there has been a demonstration of frequency comb Faraday rotation spectroscopy (not dual comb) that would be worth citing:

Johansson, Alexandra C., Jonas Westberg, Gerard Wysocki, and Aleksandra Foltynowicz. "Optical Frequency Comb Faraday Rotation Spectroscopy." *Applied Physics B* 124, no. 5 (May 1, 2018): 79. <https://doi.org/10.1007/s00340-018-6951-8>.

Reply: Magnetic rotation spectroscopy (MRS) and Faraday rotation spectroscopy (FRS) are both induced by the magnetic field, and the difference is the different physical quantities they measured. The Faraday effect causes the difference in the refractive index (Δn magnetic circular birefringence, MCB) and absorption (ΔA magnetic circular dichroism, MCD) of the left and right circularly polarized light at the resonant frequency. FRS measures the Faraday rotation angle ($\theta = \Delta n L \pi / \lambda$), where L is the distance through a medium and λ is the light wavelength. According to Eq. 16 on Page 4 of Ref. 3, the normalized FRS signal is directly proportional to the Faraday rotation angle, also known as optical rotatory dispersion (ORD). While MRS is the result of the combined action of MCB and MCD, according to Eq. 1 on Page 3 of Ref. 12. These definitions are supplemented in the **Methods** section of the revised manuscript to avoid readers confusion.

The article mentioned by the reviewer is an experimental demonstration that optical frequency comb is applied in Faraday rotation spectroscopy with the scanning Fourier-transform method. It is cited in the revised manuscript. According to the comparison in Reply 1, our technique provides an improvement in the spectral resolution, the detection sensitivity and the measurement time.

[12] Blum, F. A., Nill, K. W. & Strauss, A. J. Line shape of the Doppler-limited infrared magnetic rotation spectrum of nitric oxide. *J. Chem. Phys.* **58**, 4968-4970 (1973).

Correction1: The definition of the magnetic rotation spectroscopy (MRS) is added in the **Methods** section:

“Definition of magnetic rotation spectroscopy (MRS). The MRS is the result of the combined action of circular magnetic birefringence (MCB) and magnetic circular dichroism (MCD), which can be expressed by⁵⁶

$$I_{MRS} = 1/4 [T_+^{1/2} - T_-^{1/2}]^2 + (T_+ T_-)^{1/2} \sin^2 \theta,$$

where T_+ and T_- are the transmission spectra of left- and right-circularly-polarized light. And $\theta = \Delta n L \pi / \lambda$ is the Faraday rotation angle, where Δn is the refractive index difference of left- and right-circularly-polarized light, L is the distance through a medium and λ is the light wavelength.”

Correction2: The sentences are added to cite the article the reviewer mentioned as Ref. 23 in Paragraph 2 of the **Introduction** section:

“Recently, Aleksandra Foltynowicz’s group introduced the optical frequency comb into Faraday rotation spectroscopy for the first time, and measured the high-resolution gas-phase spectrum using a long-range scanning Fourier spectrometer²³. The sensitivity is limited to be $\sim 10^{-3}$ @3500 s by the scanning speed and intrinsic noise of the Fourier spectrometer.”

7. Fig 5. How was the absorption spectrum in (a) recorded? Is it just the same way that the reference spectrum is obtained (without purging the cell)?

Reply: Benefitting from the delicate design of our gas cell, permanent magnet can be removed without changing the light path and gas sample. Then, the crossed polarizer was rotated by 0.45° and the absorption spectrum was recorded by the same dual-comb spectrometer. The reference spectrum was obtained in the same way after purging the cell.

Correction: The sentence is added to explain the way to measure the absorption spectrum in the **Magnetic vibrational CD and ORD of nitric oxide** section:

“The crossed polarizer of the dual-comb optical activity spectrometer was rotated by 0.45° to record the absorption spectrum of NO after removing the permanent magnet.”

8. Fig 6 caption: Linear birefringence (LD) should be (LB).

Reply: The caption of Figure 6 in the article has been revised to (LB).

Correction: We replace the ***“Magnetic vibrational linear dichroism (LD) and linear birefringence (LD) of nitrogen dioxide”*** by ***“Magnetic vibrational linear dichroism (LD) and linear birefringence (LB) of nitrogen dioxide.”***

Reviewer #2 (Remarks to the Author):

The authors report on the mid-infrared dual-comb optical activity spectroscopy that enables fast measurement of chiral molecules with high spectral resolution. They

applied their technique to the molecular spectroscopy of NO₂ and NO to observe Zeeman-split absorption lines and compared their experimental results with theoretical calculations. The paper is well written, and the validity of the data is adequate. Because optical activity measurements of chiral molecules surely broaden the applicability of the dual-comb spectroscopy, I recommend to publish the article to Nature Communications after some minor revisions as described below.

Reply: Thank you for your appreciation of our article and current work, and we provide a point-by-point response to these comments along with your revision.

(1) In the Introduction, the authors refer to a time-resolved VCD spectroscopy, but they do not perform the experiment in this article. This is a confusing statement, and it is better to describe it in conclusion as a future perspective.

Reply: Thank you for your comments on the shortcomings of our article structure. We are so eager to demonstrate the potential advantages of our technology that we ignore the rationale for writing articles. To avoid the confusing statement, we delete the discussion on time-resolved VCD spectroscopy in the **Introduction** section, and this content is described at the end of the **Discussion** section.

Correction: The sentence is added to describe the time-resolved VCD spectroscopy at the end of the **Discussion** section:

“In addition, based on the femtosecond characterization of OFCs, time-resolved VCD spectroscopy with femtosecond time resolution can be realized to observe ultrafast transient dynamics, such as those occurring during protein folding, asymmetric chemical reactions and free radical dynamics.”

(2) In the principle and experimental setup. A detail of the mid-infrared optical frequency combs they used is required even though they cite their previous article (Ref. 38). At least, center wavelength, spectral bandwidth, and the output power of both combs are required.

Reply: These details has been added in the Principle and Experimental Setup of the revised manuscript.

Correction: The details are added in the **Principles and Experimental Setup** section:

“When a mid-infrared ICL centered at 3.37 μm is served as the signal of the nonlinear optical modulation process, the power of our dual-comb sources are 150 mW and 190 mW, respectively, and the spectral coverage is 3.30-3.60 μm .”

(3) Schematic figure of the experimental configuration of the NdFeB magnet is required to explain how they apply magnetic field to the gas molecule.

Reply: In the revised Fig. 1b of experimental setup, the schematic diagram that

magnetic field created by permanent magnet acts on paramagnetic molecules is added to illustrate that the gas molecules are affected by the external magnetic field in this direction.

Correction: The schematic diagram is modified to illustrate that magnetic field created by permanent magnet acts on paramagnetic molecules:

“

”

(4) How about the polarization stability of the signal comb? Instability of the polarization of the signal comb will cause the intensity fluctuation of the comb's power after passing through the polarizer.

Reply: In Ref. 1 (<https://doi.org/10.1364/PRJ.422397>), we describe the generation of optically modulated dual frequency combs in detail. Based on the optical parametric amplification process, the mid-infrared frequency combs are obtained in a quasi-phase-matched nonlinear crystal (periodically poled lithium niobate, PPLN). This process is polarization dependent, which ensures the good polarization characteristics of the mid-infrared frequency combs. We measured the long-term stability of the power before and after a high-extinction-ratio polarizer, where the stability of the power before the polarizer is dominated by power fluctuation and that after the polarizer is affected by both power fluctuations and polarization state instability. The insertion loss of the polarizer results in a decrease in the average power. As shown in Fig. R3a and R3b, the power fluctuations before and after the polarizer are at the same level, which proves that the polarization characteristics have no significant impact on the power stability of the system. Besides, benefitting from the background-free spectral measurement of the cross-polarization scheme, the power jitter will not degrade the measurement performance of the system for weak OA signals. Of course, this influence can be avoided by simultaneously measuring the signal and the reference

spectrum².

Fig. R3 **a** Power stability of the frequency comb over 30 minutes before the polarizer. **b** Power stability of the frequency comb over 30 minutes after the polarizer.

[1] Zuo, Z. et al. Broadband mid-infrared molecular spectroscopy based on passive coherent optical–optical modulated frequency combs. *Photon. Res.* **9**, 1358-1368 (2021).

[2] Coddington, I., Swann, W. C. & Newbury, N. R. Coherent dual-comb spectroscopy at high signal-to-noise ratio. *Phys. Rev. A* **82**, 043817 (2010).

(5) It seems that the denominator of the most right-hand side of Eq. (5) is not correct. It should be " $\mathfrak{F}[S_{SC,\parallel}(t)] / \sin \alpha$ ", isn't it? Please also check Eqs. (6) and (7).

Reply: We thank the reviewer for the careful reading and comment. In the revised manuscript, Eqs. (5-7) have been corrected.

Correction: Eqs. (5)-(7) are modified as

“

$$\Delta\chi(\omega) \propto \frac{\tilde{E}_{FID,\perp}(\omega)}{\tilde{E}_{SC,\parallel}(\omega)} = \frac{\tilde{E}_{FID,\perp}(\omega)\tilde{E}_{LO,\perp}^*(\omega)}{\tilde{E}_{SC,\parallel}(\omega)\tilde{E}_{LO,\perp}^*(\omega)} = \frac{\mathfrak{F}[S_{FID,\perp}(t)]}{\mathfrak{F}[S_{SC,\parallel}(t)]/\sin \alpha} \quad (5)$$

$$\Delta A = \frac{4}{2.303} \text{Im} \left(\frac{\tilde{E}_{FID,\perp}(\omega)}{\tilde{E}_{SC,\parallel}(\omega)} \right) = \frac{4}{2.303} \text{Im} \left(\frac{\mathfrak{F}[S_{FID,\perp}(t)]}{\mathfrak{F}[S_{SC,\parallel}(t)]/\sin \alpha} \right) \quad (6)$$

$$\Delta\varphi = \text{Re} \left(\frac{\tilde{E}_{FID,\perp}(\omega)}{\tilde{E}_{SC,\parallel}(\omega)} \right) = \text{Re} \left(\frac{\mathfrak{F}[S_{FID,\perp}(t)]}{\mathfrak{F}[S_{SC,\parallel}(t)]/\sin \alpha} \right) \quad (7)$$

”

(6) For the theoretical calculations in the Supplementary Information (SI). The authors calculate the MVCD and MORD spectra referring Reference S11, but with little

information. For example, according to S11, the Voigt-shape function is a function of self-collision HWHM and foreign gas collision HWHM, but the concrete values for their calculation are not indicated in the SI. The authors should describe the details of the functions they used for the theoretical calculations.

Reply: Thanks for the reviewer's suggestion. We add more calculation information in the revised Supplementary Information, including the absorption profile and the parameters used in our simulation.

Correction: The details of the theoretical calculations are added in the Supplementary Information:

"In order to reduce the computation time in full-vibrational-band simulation, the absorption profile is expressed by the pseudo-Voigt¹⁸,

$$F_{pV} = (1 - \eta)F_G(\nu; \gamma_G) + \eta F_L(\nu; \gamma_L),$$

$$F_G(\nu; \gamma_G) = 1/\pi^{1/2}\gamma_G e^{-\nu^2/\gamma_G^2} \text{ and } F_L(\nu; \gamma_L) = 1/(\pi\gamma_L)(1 + \nu^2/\gamma_L^2)^{-1},$$

where ν is the center frequency of the absorption transition, $F_G(\nu; \gamma_G)$ and $F_L(\nu; \gamma_L)$ are the Gaussian and Lorentzian functions, η is the mixing-parameter of the two functions,

$$\eta = 1.36603\gamma_L/\gamma - 0.47719(\gamma_L/\gamma)^2 + 0.11116(\gamma_L/\gamma)^3,$$

$$\gamma = (\gamma_G^5 + 2.69299\gamma_G^4\gamma_L + 2.42843\gamma_G^3\gamma_L^2 + 4.47163\gamma_G^2\gamma_L^3 + 0.07842\gamma_G\gamma_L^4 + \gamma_L^5)^{1/5}.$$

The profile parameters and the measurement parameters used in the molecular simulations are shown in the following Table S4.

Table S4. Parameters used in the simulation.

	γ_D/cm^{-1}	γ_L/cm^{-1}	$P/mbar$	T/K	L/cm
NO₂ for Faraday effect	$9.11 * 10^{-7}\nu_0^*$	0.005	8	298	7.6
NO₂ for Voigt effect	$9.11 * 10^{-7}\nu_0^*$	0.0023	1.7	298	30
NO	$11.28 * 10^{-7}\nu_0^*$	0.012	65	298	7.6

* ν_0 is the wavenumber of the transition line.

And due to the mode-resolved characteristics of our dual-comb spectrometer, no instrumental lineshape was taken into account¹⁹. The MVCD (ΔA) is obtained by calculating the difference in absorbance between σ^\pm ,

$$\Delta A = -lg T_+(v) + lg T_-(v) = -lg \frac{T_+(v)}{T_-(v)}.$$

According to the Kramers-Kronig relationship, the phase spectra of the Zeeman transitions could be calculated by²⁰

$$\varphi_\pm(v) = \frac{2}{\pi} \int_0^{+\infty} \frac{v' \ln T_\pm(v')}{v'^2 - v^2} dv'.$$

And the MORD ($\Delta\varphi$) is expressed by

$$\Delta\varphi = \frac{\varphi_+(v) - \varphi_-(v)}{2}.$$

Reviewer #3 (Remarks to the Author):

This manuscript demonstrates the detection of magnetic field induced optical activity using a dual-comb approach. The approach is a rather straightforward modification of standard dual-comb spectroscopy, namely the LO comb has linear polarization that is orthogonal to that of the signal comb. The method is demonstrated on a gas of nitrogen dioxide that is placed in a magnetic field.

The results appear to be valid and the manuscript is generally well written, although it could use review/copy editing by a native English speaker to help scrub some minor issues with language usage, although none of them interfere with understanding the technical content.

Although the results appear correct and valid, the authors do not address the need for this advance nor the impact it may have. Generic statements are made about improving the spectral resolution and sensitivity, which are all reasonable. However, it is not clear that there are any situations where these improvements would be groundbreaking and enable the discovery of new science, or be technologically relevant.

Given that it reports an incremental improvement on the detection of optical activity and similarly is a variant on standard dual comb spectroscopy, I do not see that this manuscript has the novelty or impact required for publication in Nature Communications.

Reply: We thank the reviewer for the comments.

Firstly, we applied the dual-comb technique in optical activity spectroscopy (OAS) for gas-phase molecules for the first time, achieving hundred-MHz resolution and sub-millisecond level measurement speed, which set new records in gas-phase OAS study. This technique provides a powerful platform for gas-phase optical activity, which will be used in the fine-structure analysis and calibrating ab-initio theoretical predictions. For example, in this paper, the high-resolution and high-sensitivity DC-OAS technique has become a powerful tool for exploring the fine-level structure of free radicals, revealing spin-rotation coupling in nitrogen dioxide by the rotationally resolved MVCD spectrum.

Secondly, to demonstrate the diverse application potential of this technology, we complemented the measurement experiment for the optical activity of chiral limonene. (See the **Vibrational optical activity spectra of chiral limonenes in liquid-phase system** section of the revised manuscript for more details.) In the revised manuscript, it's shown that the system is capable of detecting the optical-activity signal of limonene as weak as 10^{-5} in several-second sampling time, whereas measurements with the commercial VCD spectrometer required multiple hours to obtain such sensitivity. Accordingly, we believe that our technique presents significant advances in rapid measurement of weak optical activity and is of great potential in time-resolved VCD spectroscopy.

Correction1: The **Abstract** and **Introduction** sections are modified to clearly illustrate

the improvements of DC-OAS.

Correction2: The **Vibrational optical activity spectra of chiral limonenes in liquid-phase system** section is added to introduce the preliminary work in liquid-phase chiroptical activity.

“

Fig. 7 Vibrational optical activity signal of chiral limonenes. **a**, The signal interferogram (black line, after averaging over 1 hour) and reference interferogram (red line, after averaging over 5 minutes) through the R-limonene sample within 2.5 μs. The signal interferogram is magnified 150 times for comparison. The chiral limonenes dissolved in CCl₄ with the analyte concentration of 110 mM and the path length of 1 mm. Inset: zoom-in plot of the interferograms. **b** and **c**, Absolute VCD (ΔA) and ORD (Δφ) spectra measured by DC-OAS with the samples of R-limonene (blue line) and S-limonene (red line) and their 1:1 racemic solution (pink line).

Vibrational optical activity spectra of chiral limonenes in liquid-phase system. In order to demonstrate the ability of our scheme with chiral optical activity measurement, we provided the proof-of-principle results on the (R)- and (S)-limonene and their racemic mixture, where these samples are dissolved by CCl₄ to the concentration of 110 mM. This experiment is performed without any other modification except that the repetition frequency difference is set to 600 Hz. In this case, the period of a single interferogram is about 1.67 ms ($1/f_r$), where only 2.5-μs-long raw data is sampled. Figure 7a shows the measured signal and reference interferograms, where the apodized resolution is 2.4 cm⁻¹. The measurement for each kind of limonenes lasts sixty minutes, which is limited by the refresh rate of the interferograms. In fact, the total sampling time of the available interferograms is only 5.4 s. The low duty cycle is caused by the relatively low repetition rate of our dual-comb source, which can be overcome by the high-repetition-rate frequency combs³¹. Figure 7b and 7c compare VCD

and ORD spectra of the R- (blue line) and S-limonene (pink line) and their racemic mixture (red line), respectively. The baseline from the sample cell has been removed by measuring the reference spectra of the pure CCl₄. As expected, the VCD and ORD signals of R- and S-limonene are nearly identical but their signs opposite to each other, while the optical activity signals of the racemic mixture does not exhibit observable structure. These results demonstrate the capability of our technique to measure chiroptical activity as weak as 10⁻⁵ and characterize chiral species. These results further demonstrate the potential of our technique to rapidly measure weak chiroptical activity signals, which is expected to advance time-resolved VCD spectroscopy to study the secondary structure dynamics of bio-chiral molecules.”

REVIEWERS' COMMENTS

Reviewer #1 (Remarks to the Author):

Thank you for the thorough response and edits. I recommend publication.

One comment is that the resolution of the figures is low in the pdf version that I saw. I did not check the raw figure files, but please make sure the resolution is high in the final version.

Reviewer #2 (Remarks to the Author):

The authors appropriately answered my concerns, and I recommend to publish this article in Nature Communications.

Manuscript ID: NCOMMS-22-27570A;

Title: Dual-comb optical activity spectroscopy for the analysis of vibrational optical activity induced by external magnetic field;

Correspondence Authors: Dr. Wenxue Li.

Response to Reviewers

Thanks for the reviewer's comments and suggestions on our manuscript (Manuscript ID: NCOMMS-22-27570A). We reviewed the questions and suggestions carefully, and provided the point-by-point response.

Note that reviewers' comments are marked in black, and authors' responses are in blue.

Reviewer #1 (Remarks to the Author):

Thank you for the thorough response and edits. I recommend publication.

One comment is that the resolution of the figures is low in the pdf version that I saw. I did not check the raw figure files, but please make sure the resolution is high in the final version.

Reply: We are glad that the reviewer is satisfied with our revision. In the revised manuscript and **Response to Reviewers**, we provided the vector images, which can still maintain a considerable resolution after being converted to PDF version.

Correction: The vector images are provided for a considerable resolution,

Fig. 1 Basic principles and experimental setup of the DC-OAS system. **a** Sketch of the basic principles of the magnetic optical activity (MOA) response of the frequency comb in the cross-polarization scheme. The signal comb (SC) polarized along z axis, is incident on the sample cell in the magnetic field. For the Zeeman-split absorption lines of paramagnetic molecules, the left- and right-circularly polarized components (LCP: $\Delta M_j = -1$; RCP: $\Delta M_j = +1$) of the linearly polarized field have different wavelength-dependent complex propagation constants, which results in magnetic optical rotatory dispersion (MORD) and magnetic vibrational circular dichroism (MVCD). After interacting with the paramagnetic sample, the polarized direction of the spectral components involved in the resonant transition rotates, resulting in MOA signal polarized along x axis. The MOA signal in optical frequency is down-converted to radio frequency by multiheterodyne beating with the local oscillator (LO) comb. The linear polarizer (LP) oriented along x axis is used to filter out the non-MOA spectral components. **b** Schematic of the DC-OAS system. The ultrashort mid-infrared pulse emitted from the SC passes through a half-wave plate ($\lambda/2$) and linear polarizer LP1 to ensure linear polarization. Through interaction with paramagnetic molecules in the magnetic field of the NdFeB magnet, vertical components of the electric field are created as free-induced decay (FID). Only the MOA FID response is extracted by linear polarizer LP2, and is combined with the beam from the LO comb by a beam splitter (BS). The repetition rates between two combs are slightly different to obtain a dual-comb asynchronous optical sampling of the electric response of MOA FID. After an optical filter (F) to avoid aliasing, the interferogram signals are focused on the photodetector (PD) by a lens (L) and converted into electrical signals, which are recorded digitally by a data-acquisition card (Alazar Tech, ATS9350).

Fig. 2 Experimental interferograms and mode-resolved DC-OAS spectra. **a** Comparison between the unaveraged and 1200-average interferograms within a single period of 833 μs ; the inset plot is the induced-free decay of magnetic optical activity (MOA) within 14 μs . **b** Spectra of the MOA signal for 833 μs (grey line) and 1 s (blue line) measurement times. **c** Partial spectra of **b** ranging from 2901 cm^{-1} to 2904 cm^{-1} of the MOA. **d** Mode-resolved MOA spectrum of nitrogen dioxide with a recording time of 1 s. The spectral range from 87.08 THz to 87.10 THz is shown to reveal the series lines of the ${}^0\text{Q}(K_a=4)$ subbranch; the blue curve is the contour fitting. **e** Line of the ${}^0\text{Q}(7_{4,3})$ transition is composed of several combs with a 108.4 MHz spacing. **f** the single comb with 90 kHz linewidth.

Fig. 3 Measured intensity spectrum of DC-OAS. a Experimental DC-OAS intensity spectrum of NO₂ obtained from a continuous 1000 s recording and sampled at exactly the comb-line spacing of 108 MHz from 2870 to 2940 cm⁻¹. The spectrum is normalized by a background spectrum, removing the influence of spectral shape on the relative intensity of spectral lines. **b, c, d, e** and **f** Magnified views surrounded by rectangles in **a**. **b, c** and **d** Series lines from the ²¹P(K_a=4), ²⁰Q(K_a=4), ²¹R(K_a=4) sub-branch in the $\nu_1 + \nu_3$ band with N=K_a-21. **e** Series lines of the ²¹Q(K_a=8) subbranch in cold $\nu_1 + \nu_3$ band (top) and the magnified weak spectral lines (bottom). **f** Measured intensity spectrum of DC-OAS in the $\nu_1 + \nu_2 + \nu_3 - \nu_2$ hot band.

Fig. 4 Magnetic vibrational optical activity signals of nitrogen dioxide. **a** and **b** Retrieved MVCD and MORD spectra of NO_2 covered from 2870 to 2940 cm^{-1} , respectively. **c** and **d** MVCD and MORD spectral lines of the ${}^0Q(K_a=4)$ subbranch, the dashed lines below are the simulation results. **e**, **g** and **i** (f, **h** and **j**) MVCD (MORD) lines of the ${}^0P(14_{5,9})$, ${}^0P(30_{0,30})$ and ${}^0P(34_{6,28})$ transitions. The transition frequency of the unresolved doublets (labelled -1/2 and 1/2) are marked by pink dotted and solid lines, respectively. The MVCD lineshape of the “-1/2 1/2” doublet is presented as “V-type”, while the lineshape of the “1/2 -1/2” doublets are “ Λ -type”. The dashed lines in the above figures are the simulated results.

Fig. 5 Measured magnetic vibrational optical activity signals of nitric oxide with DC-OAS in atmospheric environment. **a** and **b** Comparison of the absorption spectrum and MOA intensity spectrum of nitric oxide. The dips with red curves represent the absorption lines of NO. **c** and **d** MVCD and MORD spectra measured by DC-OAS. Grey shaded area represents the frequency area masked by H_2O saturated absorption, and missing spectral lines are completed with calculated data (magenta dashed lines). The subplots of **e**, **g** and **i**, **h** and **j** depict partial MVCD (MORD) lines from the P-branch, Q-branch and R-branch, and the dashed lines in the above figures are the simulated results.

Fig. 6 Magnetic vibrational linear dichroism (LD) and linear birefringence (LB) of nitrogen dioxide. **a** Differential absorbance ΔA between orthogonally polarized light ($\Delta A = A_{\pi} - A_{\sigma}$); **b** Differential phase $\Delta\varphi$ between orthogonally polarized light ($\Delta\varphi = \frac{\pi L}{\lambda}(n_{\pi} - n_{\sigma})$). **c** **(d)** LD (LB) lines of the $^{10}\text{Q} (K_a=3)$ subbranch with $N=3-9$; **e** **(f)** LD (LB) lines of the $^{10}\text{Q} (K_a=5)$ subbranch with $N=5-13$. The dashed lines below is the simulated results.

Fig. 7 Vibrational optical activity signal of chiral limonenes. a

The signal interferogram (black line, after averaging over 1 hour) and reference interferogram (red line, after averaging over 5 minutes) through the R-limonene sample within 2.5 μs . The signal interferogram is magnified 150 times for comparison. The chiral limonenes dissolved in CCl_4 with the analyte concentration of 110 mM and the path length of 1 mm. Inset: zoom-in plot of the interferograms. **b** and **c** Absolute VCD (ΔA) and ORD ($\Delta \phi$) spectra measured by DC-OAS with the samples of R-limonene (blue line) and S-limonene (red line) and their 1:1 racemic solution (pink line).

Reviewer #2 (Remarks to the Author):

The authors appropriately answered my concerns, and I recommend to publish this article in Nature Communications.

Reply: We are glad that the reviewer is satisfied with our revision.